# The CPLANE protein Intu protects kidneys from ischemia-reperfusion injury by targeting STAT1 for degradation

Shixuan Wang[1], Aimin Liu [2], Guangyu Wu[3], Han-Fei Ding[4], Shuang Huang[5], Stanley Nahman[6] & Zheng Dong[1,7]

Intu is known as a ciliogenesis and planar polarity effector (CPLANE) protein. Although roles for Intu have been reported during embryonic development and in the context of developmental disorders, its function and regulation in adult tissues remain poorly understood. Here we show that ablation of Intu specifically in kidney proximal tubules aggravates renal ischemia-reperfusion injury, and leads to defective post-injury ciliogenesis. We identify signal transducer and activator of transcription 1 (STAT1) as a novel interacting partner of Intu. In vitro, Intu and STAT1 colocalize at the centriole/basal body area, and Intu promotes proteasomal degradation of STAT1. During cell stress, Intu expression preserves cilia length and cell viability, and these actions are antagonized by STAT1 expression. Thus, we propose a role for Intu in protecting cells and tissues after injury by targeting STAT1 for degradation and maintaining primary cilia.

[1] Department of Cellular Biology and Anatomy, Medical College of Georgia at Augusta University and Charlie Norwood VA Medical Center, Augusta, GA 30912, USA. [2] Department of Biology, Eberly College of Sciences, Huck Institute of the Life Sciences, The Pennsylvania State University, University Park, PA 16802, USA. [3] Department of Pharmacology and Toxicology, Medical College of Georgia at Augusta University, Augusta, GA 30912, USA. [4] Cancer Center and Department of Pathology, Medical College of Georgia at Augusta University, Augusta, GA 30912, USA. [5] Department of Anatomy and Cell Biology, University of Florida College of Medicine, Gainesville, FL 32610, USA. [6] Department of Medicine, Medical College of Georgia at Augusta University, Augusta, GA 30912, USA. [7] Department of Nephrology and Institute of Nephrology, Second Xiangya Hospital, Central South University, Changsha, Hunan 410011, China. Correspondence and requests for materials should be addressed to Z.D. (email: zdong@augusta.edu)

Planar cell polarity (PCP) refers to the coordinated alignment of cell polarity across the tissue plane, which is essential for embryonic development and normal tissue function in animals[1–7]. PCP is established and maintained by the complex machinery of two modules (PCP core and Fat systems) and effector proteins[1,7]. The effector proteins are further divided into two groups with Daam, Rac and Rho in group 1, and Intu, Fuzzy and Fritz/Wdpcp in group 2. Recently, a potential relationship between PCP and primary cilia has been suggested[6,8,9]. In this regard, several PCP proteins have been localized at the cilium or basal body area[10–13], and dysfunction of these PCP proteins impair ciliogenesis, causing cilia-associated diseases called ciliopathies[14–18]. This is well-exemplified by the PCP effector protein Intu, which accumulates at the base of cilia or basal body in *Drosophila* for the recruitment of intraflagellar transport proteins and the regulation of the subapical actin network for ciliogenesis[10,11]. Disruption of *Xint* (orthologue of *Intu*) in *Xenopus* embryos led to defects in cilia and neural tube closure[19]. Recent work further discovered several *Intu* mutations in ciliopathy patients[11]. Based on their dual functions, Intu and related PCP proteins are classified as CPLANE (ciliogenesis and planar polarity effector) proteins[11].

Almost every kidney tubular cell has a primary cilium protruding toward the lumen. Dysregulation of several cilia and PCP proteins has been implicated in the pathogenesis of renal diseases, such as polycystic kidney disease[20]. Moreover, ciliary defects sensitize kidneys to ischemia-reperfusion injury (IRI)[21,22]. In addition, following renal IRI, there seems to be an adaptive growth of primary cilia in kidney proximal tubules[23–26]. While these findings suggest a role of primary cilia and probably PCP in kidney injury and repair, the underlying mechanism is poorly understood.

In this study, we localize endogenous Intu at the centriole/basal body area in mouse and rat kidney proximal tubular cells. Functionally, we find that knockout of *Intu* specifically in kidney proximal tubular cells aggravates ischemic kidney injury in mice. Mechanistically, we show that Intu likely interacts with STAT1 at centriole/basal body area to induce proteasomal degradation of STAT1. Thus, Intu may protect cells by interacting with STAT1 to induce its degradation.

## Results

**Knockout of *Intu* from proximal tubules aggravates IRI.** We generated kidney proximal tubule-specific *Intu* knockout (PT-Intu-KO) and wild-type (PT-Intu-WT) mice by crossing PEPCK-Cre mice with Intu-floxed mice[27,28] (Fig. 1a). Intu expression in kidney tissues from both KO mice and WT mice

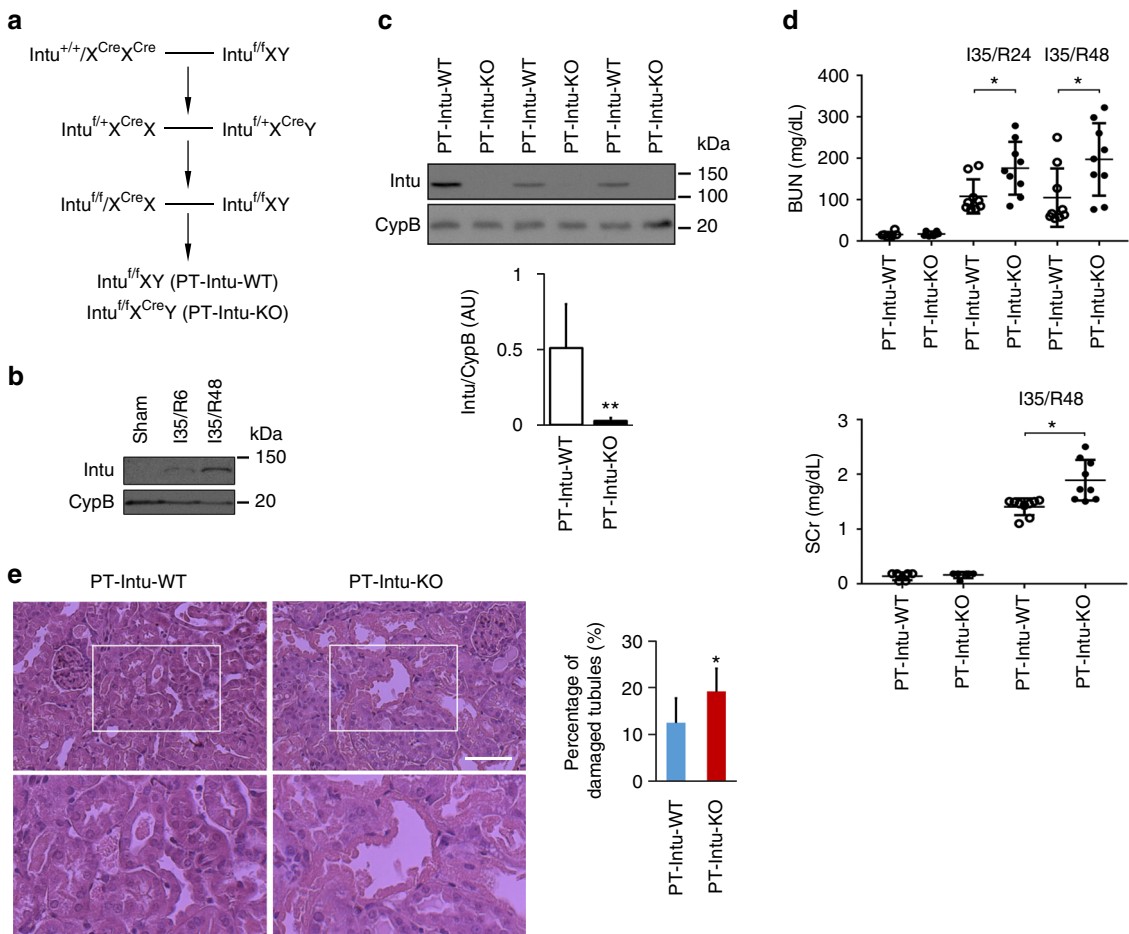

**Fig. 1** Knockout of *Intu* from kidney proximal tubular cells aggravates renal IRI in mice. **a** Breeding protocol for generating PT-Intu-KO mouse model. **b** Induction of Intu in mouse kidneys by 35 min of bilateral renal ischemia with 6 or 48 h of reperfusion. **c** Intu expression at renal IRI (48 h) was abolished in PT-Intu-KO mice as compared to PT-Intu-WT mice (n = 3). **d** PT-Intu-KO mice and PT-Intu-WT mice had similarly low levels of BUN and SCr under control condition (n = 9), but renal IRI induced higher BUN and SCr in PT-Intu-KO mice (n = 9). **e** Renal IRI induced more severe renal tubular damage in PT-Intu-KO mice than in PT-Intu-WT mice as shown by representative histology and quantification of the percentage of damaged tubules (n = 6). Quantitative data are mean ± s.d. (error bars). Paired or grouped t test was used. *p < 0.05, **p < 0.01. Scale bar: 100 μm. AU arbitrary unit

was low and, upon renal IRI, Intu was induced in WT kidneys but not in KO kidneys (Fig. 1b, c; Supplementary Fig. 1, 2), validating the knockout model. Under control conditions, both KO and WT mice had normal renal function as shown by low levels of blood urea nitrogen (BUN) and serum creatinine (SCr). Upon renal IRI, KO mice had significantly higher BUN and SCr than WT mice at both 24 and 48 h of reperfusion (Fig. 1d). Consistently, KO mice showed higher degrees of renal tubule damage (Fig. 1e).

**Intu regulates kidney tubular cell death**. To study the role of Intu in cell death regulation, we performed Terminal deoxynucleotidyl transferase dUTP nick end labeling (TUNEL) assay on kidney sections. TUNEL staining revealed more death cells in KO mouse kidney tissues (Fig. 2a, Supplementary Fig. 3). To further functionally characterize Intu, we examined the effect of stable Intu knockdown with specific short hairpin RNA (shRNA) in BUMPT cells (shIntu; Fig. 2b, c). Upon azide or CCCP (carbonyl cyanide 3-chlorophenylhydrazone) treatment, Intu-knockdown cells had significantly higher levels of cell death than control shRNA-transfected cells (shControl; Fig. 2d, e). We further established tetracycline-inducible Intu expression cells,

which expressed Intu upon doxycycline treatment (Fig. 3a: D-Rex Intu3, 5, 10). Induced Intu expression protected renal tubular cells from azide-induced metabolic stress and cisplatin-induced toxic injury morphologically (Fig. 3b, c); consistently Caspase-3 and Parp cleavage in these cells was also suppressed (Fig. 3d).

**STAT1 is a novel Intu-interacting protein**. To understand how Intu protects, we probed its interacting proteins by tandem affinity purification (TAP) assay, which pulled down at least seven distinctive bands on silver-stained SDS-PAGE gel (Supplementary Fig. 4). Liquid-chromatography tandem mass spectrometry (LC-MS/MS) analysis of these bands suggested the presence of 160 proteins (Supplementary Table 1). Among them, two peptides of STAT1 (Fig. 4a) were identified suggesting that STAT1 might be a novel interacting partner of Intu. We further confirmed Intu/STAT1 interaction by streptavidin (STV)-pulldown assay and co-immunoprecipitation (IP) (Fig. 4b, c). The interaction of endogenous Intu and STAT1 was further validated in mouse kidneys (Fig. 4d). Moreover, in STV-pulldown assay, the Intu N-terminus containing PDZ domain precipitated endogenous STAT1, whereas Intu mutants lacking this domain did not,

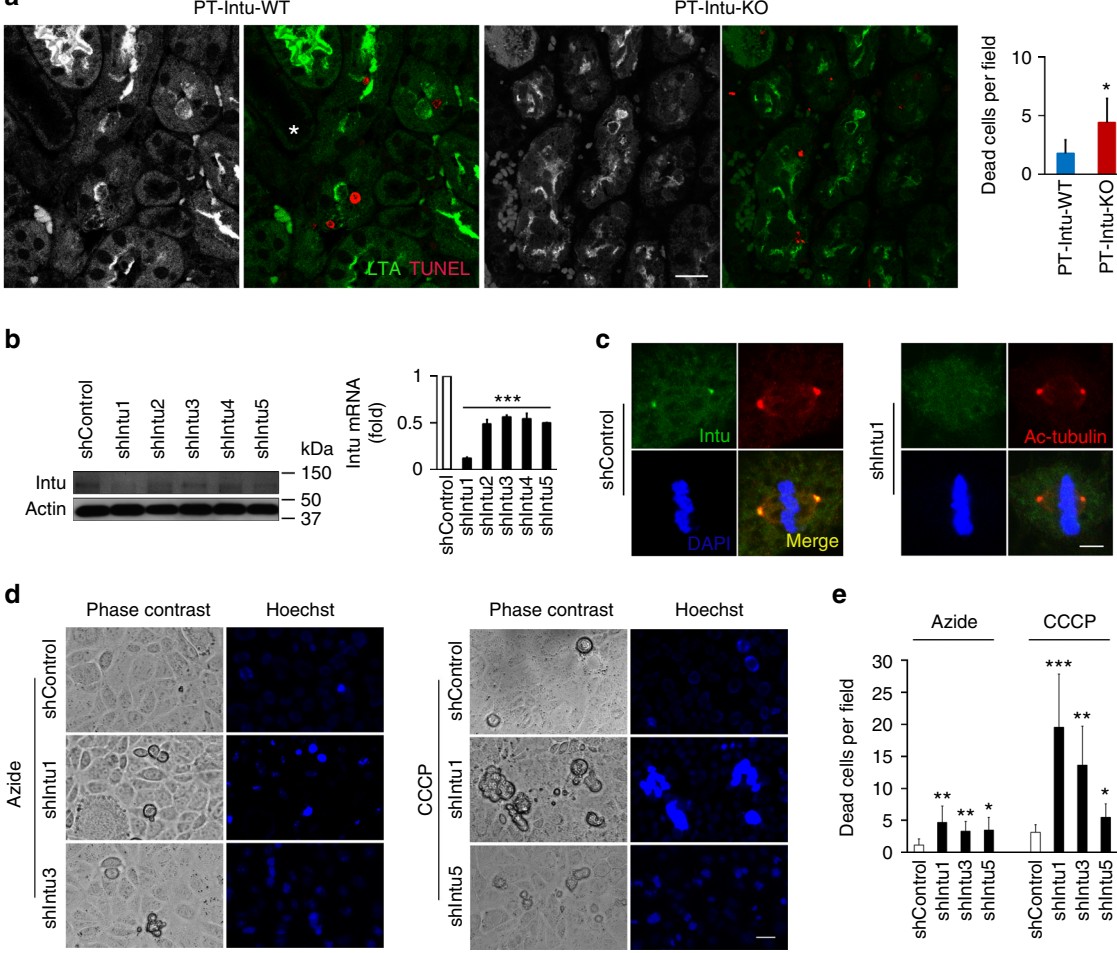

**Fig. 2** Suppression of Intu increases cell death during renal IRI injury in vivo and metabolic stress in vitro. **a** TUNEL assay showing more death cells (red) in proximal tubules (green by LTA staining) of PT-Intu-KO kidney tissues following renal IRI than PT-Intu-WT ($n = 8$). Please note that LTA staining pattern was disrupted due to injury. The asterisk indicates a likely non-PT tubule with no LTA staining and different morphology. **b**, **c** Establishment of stable Intu-knockdown BUMPT cells by transfecting Intu shRNAs (shIntu) confirmed by immunoblot, qRT PCR, and immunostaining. **d** Representative cell and nuclear images showing more cell death in Intu-knockdown cells after azide or CCCP treatment. Two knockdown clones targeting different sites were presented. **e** Quantification of cell death in Intu-knockdown cells and control shRNA-transfected cells after azide or CCCP treatment. Quantitative data are mean ± s.d. (error bars). Paired or grouped $t$ test was used. *$p < 0.05$, **$p < 0.01$, ***$p < 0.001$. Cell nuclei were stained with DAPI. Scale bar (μm): 20 (**a**, **d**), 10 (**c**)

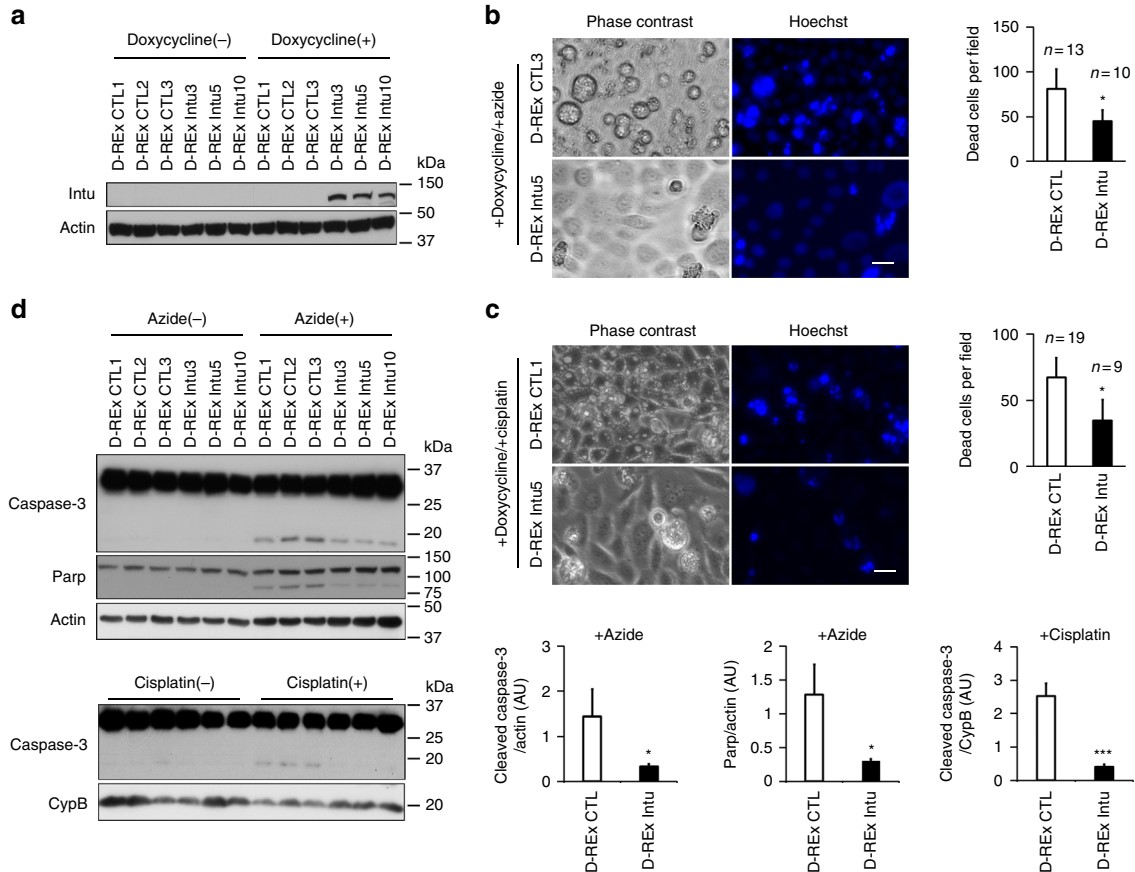

**Fig. 3** Inducible Intu expression suppresses cell death. **a** Inducible Intu expression in RPTC cells. Three inducible cell clones (D-REx Intu3, 5, 10) were treated with doxycycline to induce Intu expression shown by immunoblotting in comparison to three empty vector controls (D-REx CTL1, 2, 3). **b** Less cell death was observed in Intu-expressing cells after azide treatment. **c** Resistance of Intu-expressing cells to cisplatin treatment. **d** Immunoblot analysis of Caspase-3 and Parp cleavage to confirm less apoptosis in Intu-expressing cells after azide or cisplatin treatment. Quantitative data are mean ± s.d. (error bars). Grouped *t* test was used.*$p < 0.05$, ***$p < 0.001$. Scale bar: 20 μm

suggesting a pivotal role of the PDZ domain in mediating Intu/STAT1 interaction (Fig. 4e). To study what modulates the interaction between Intu and STAT1, we treated cells with interferon(IFN)-γ, azide, or cisplatin, three inducers of STAT1 activation and phosphorylation. It was found that all three treatments increased Intu-bound STAT1 (Fig. 4f). We then determined where inside the cell Intu and STAT1 interact. Our immunofluorescence (IF) analysis detected Intu accumulation in the basal body/centriole area labeled by γ-tubulin during both interphase and mitosis in renal tubular cells and 293FT cells (Fig. 4g, Supplementary Fig. 5a). We further revealed primary cilia by IF of acetylated (Ac)-tubulin, and found that Intu was located at the base of primary cilia, i.e., basal body (Fig. 4h). Such staining pattern of Intu was also confirmed with a published Intu antibody 1155PB4[27] (Supplementary Fig. 5c). Likewise, transfected CBP-SBP-Intu trafficked to the basal body/centriole area in 293FT cells (Supplementary Fig. 5b), being consistent with the observation in *Xenopus* multiciliated cells[10,20]. For STAT1, we observed immunostaining of total STAT1 around the γ-tubulin-labeled basal body/centriole area, while phospho-STAT1$^{S727}$, though less abundant, appeared much more concentrated at the centriole (Fig. 4i). Thus, basal body/centriole appears to be one interacting site of Intu and STAT1.

**Intu targets STAT1 for degradation**. Why does Intu interact with STAT1 likely at the centriole or basal body? To address this,

we first examined the effect of Intu on STAT1 expression. When co-transfected, Intu reduced the expression of both total STAT1 and its phosphorylation mutants (STAT1αY701F, STAT1αS727A) (Fig. 5a). Moreover, Intu suppressed the expression of both total STAT1 and phospho-STAT1$^{S727}$ during IFN-γ, azide, or cisplatin treatment in different types of cells (Fig. 5b, Supplementary Fig. 6). We further found that full-length Intu and IntuΔ268–942, but not IntuΔ1-267 or IntuΔ181-942, could suppress STAT1 expression, indicating that the N-terminal 267 aa with the PDZ domain was essential for the suppressive effect of Intu on STAT1 (Fig. 5c). In addition, the proteosome inhibitor MG132 could restore the expression level of STAT1 in the presence of Intu (Fig. 5d), suggesting that Intu may target STAT1 for degradation through the ubiquitin-proteasome system. Intu-KO mouse kidneys showed higher total and phospho-STAT1S727 than WT tissues after renal IRI (Fig. 5e), providing in vivo evidence for Intu-mediated STAT1 degradation. Consistent with kidney injury analysis (Fig. 1), Intu-KO mouse kidneys had higher levels of Caspase-3 and Parp cleavage, but lower levels of Bcl-2 and Bcl-xL, indicative of apoptosis (Fig. 5e). Together, these results suggest that Intu may suppress STAT1 to protect kidney cells and tissues.

**Intu inhibits cell death through STAT1**. To further determine their functional interaction, we co-transfected STAT1 into inducible Intu-expressing cells. In the cells with doxycycline-

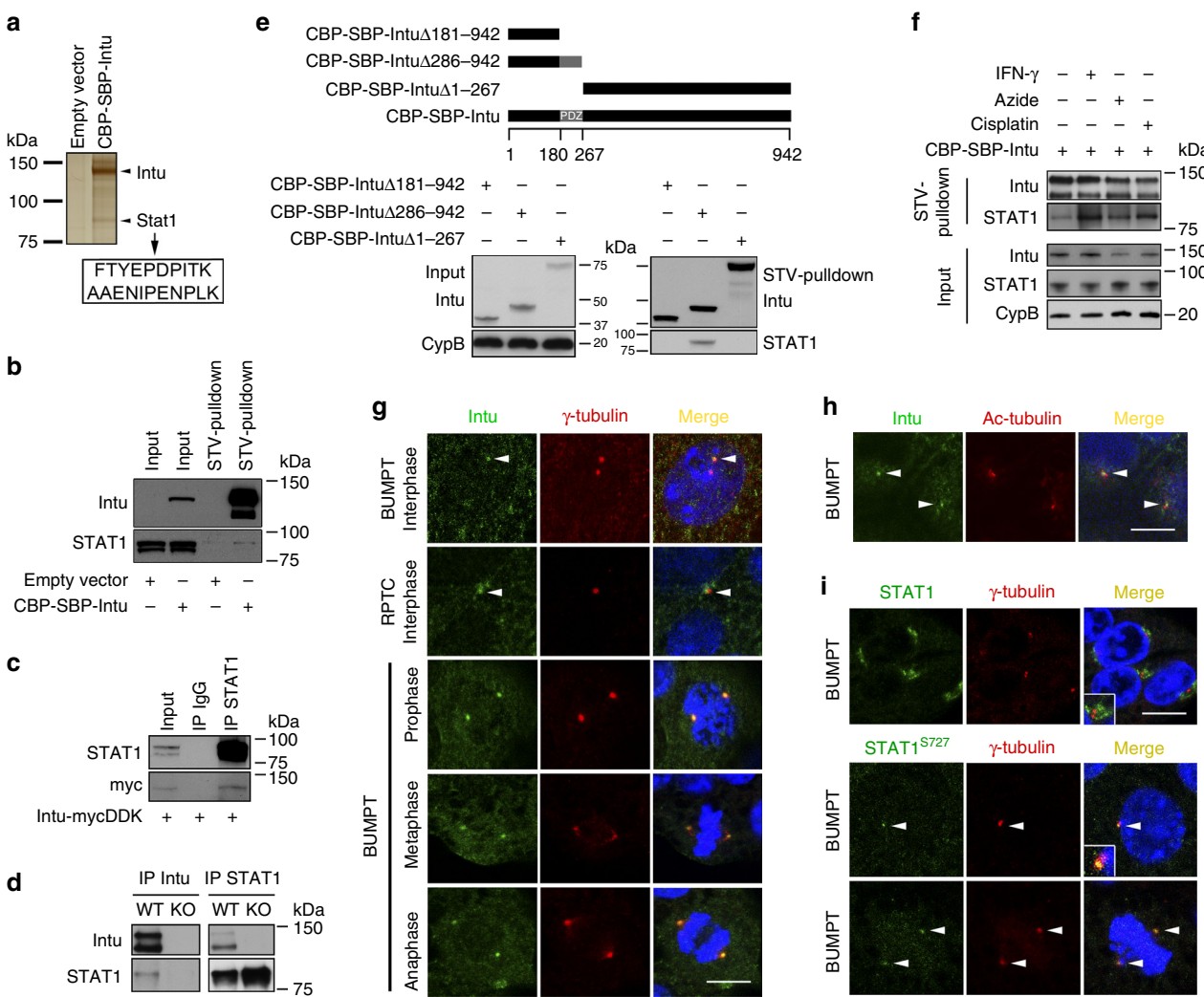

**Fig. 4** STAT1 is a novel Intu-interacting protein at the basal body/centriole area. **a** To identify Intu-interacting proteins, Intu was transiently expressed in BUMPT cells, followed by Intu pulldown by the TAP assay. Eluted proteins from the TAP were subjected to SDS-PAGE and silver staining. One band at ~90 kDa was analyzed by mass spectrometry identifying two peptides of STAT1 (in arrowed box). **b** Intu constructs (or empty vector) were transiently transfected into BUMPT cells and STV-pulldown assay was performed to determine the co-precipitation of Intu and STAT1. **c** Intu-mycDDK was transfected into 293FT cells to collect lysate for immunoprecipitation of STAT1 with anti-STAT1 antibody or non-immune IgG (control). The precipitates were then analyzed for STAT1 and myc-tagged Intu to show their co-precipitation or interaction. **d** Co-immunoprecipitation of Intu and STAT1 in kidney tissues of PT-Intu-KO or WT mice after renal IRI. **e** Intu deletion mutants were transfected into BUMPT cells. After STV-pulldown, the Intu N-terminus containing PDZ domain (CBP-SBP-Intu Δ268-942) precipitated endogenous STAT1 while the mutants lacking this domain did not. **f** Intu and STAT1 interaction during IFN-γ, azide, and cisplatin treatment. 293FT cells were transfected with Intu constructs (CBP-SBP-Intu) and subjected to different treatments. Cell lysate was collected for STV-pulldown assay to show that more STAT1 was pulled down by Intu in treated 293FT cells. **g** Double immunofluorescence of Intu and γ-tubulin (marker of basal body/centriole) showing their co-localization in cultured mouse (BUMPT) and rat (RPTC) kidney proximal epithelial cells at different cell cycle phases. **h** Double immunofluorescence of Intu and Ac-tubulin (marker of cilia) showing the localization of Intu at the base of primary cilia. **i** Double immunofluorescence of Phospho-STAT1$^{S727}$ and γ-tubulin revealed their co-localization, while double immunofluorescence of total STAT1 and γ-tubulin showed the distribution of total STAT1 around basal body/centrioles. In three panels **g–i**, cells were also stained with DAPI to show nuclear morphology in merged images. Arrow-head: basal body/centriole. Scale bar: 10 μm

induced Intu expression, Caspase-3 cleavage was attenuated after azide or cisplatin treatment (Fig. 6a). Consistently, these cells showed less cell death in morphology (Fig. 6b, c). Knockdown of STAT1 suppressed cell death under these conditions, supporting a pro-death role of STAT1 (Supplementary Fig. 8). We further knocked down STAT1 in *Intu*-silenced cells to determine the effect on cell injury (Fig. 6d). *Intu*-silenced cells were sensitive to azide- and cisplatin-induced cell death as shown by Caspase-3 cleavage/activation and apoptotic morphology, which was suppressed by STAT1 knockdown (Fig. 6e, f, g), suggesting that Intu may regulate cell injury via STAT1.

**STAT1 is involved in Intu-modulated ciliogenesis.** As a typical CPLANE protein, Intu is known to contribute to ciliogenesis[27]. After demonstrating the physical and functional interactions between Intu and STAT1 during cell stress, we wondered if STAT1 participates in Intu-mediated cilium length regulation. We initially analyzed the association of apoptosis and cilium length in kidney tubules after renal IRI in WT mice, showing that cilia grew longer in TUNEL-negative/non-apoptotic tubules, and diminished in TUNEL-positive/apoptotic tubules (Fig. 7a, Supplementary Fig. 9). We further examined primary cilia in kidney proximal tubules in PT-Intu-KO and -WT mice. Proximal

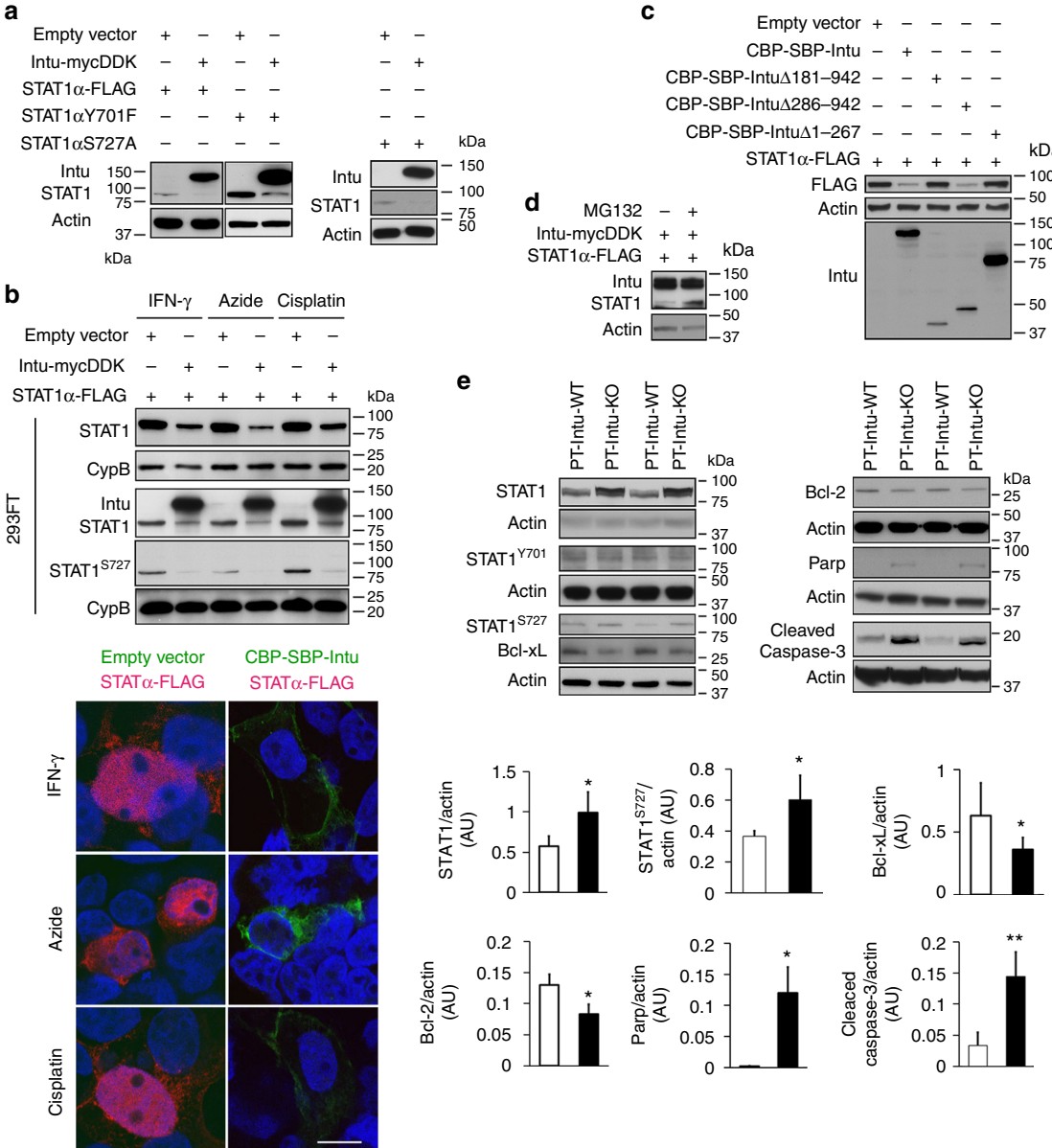

**Fig. 5** Intu targets STAT1 for degradation. **a** Co-transfection of Intu-mycDDK reduced the expression of FLAG-tagged STAT1α and its phosphorylation mutants (STAT1αY701F, STAT1αS727A) in 293FT cells. **b** Intu suppressed the expression of total and phosphorylated STAT1 during IFN-γ, azide, or cisplatin treatment of 293FT cells. **c** Intu or its deletion mutants were co-transfected with STAT1-FLAG into 293FT cells, showing the suppressive effect of full-length Intu and Intu Δ268-942 on STAT1 expression. **d** MG132 improved STAT1 expression in Intu-transfected cells. **e** Higher levels of total and phosphorylated STAT1, cleaved Caspase-3, and Parp, and lower levels of Bcl-2 and Bcl-xL were observed in PT-Intu-KO kidney tissues, in comparison to the wild type. Quantification was done by densitometry of immunoblots ($n = 3$-4). Quantitative data are mean ± s.d. (error bars). Paired $t$ test was used. *$p < 0.05$, **$p < 0.01$. Scale bar: 10 μm

tubules were labeled by lotus tetragonolobus agglutinin (LTA) or phaseolus vulgaris erythroagglutinin (PHA-E), and cilia by Ac-tubulin staining. Under sham control condition, KO and WT mice had similar cilia in proximal tubules. However, after renal IRI, WT mice had a significant growth in primary cilia of proximal tubules, whereas KO mice did not (Fig. 7b, Supplementary Fig. 10). We also determined the effects of Intu and STAT1 on primary cilia in cultured renal tubular cells. As expected, knockdown of Intu led to the shortening of primary cilia under control conditions, while inducible overexpression of Intu could preserve cilia during cell injury (Fig. 7c, d). In sharp contrast, knockdown of STAT1 resulted in longer cilia and higher percentage of cells with cilia (Fig. 7e). Moreover, STAT1 knockdown

partially restored cilium length in *Intu*-silenced cells (Fig. 7f). Conversely, induced Intu expression led to the attenuation of STAT1 activation (phospho-STAT1$^{S727}$) at the basal body/centriole area (Fig. 7g). These results suggest a reciprocal regulation of primary cilia by Intu and STAT1.

## Discussion

In this study, we have demonstrated a protective role of the CPLANE protein Intu in renal IRI in vivo and toxic/metabolic stress in vitro. Mechanistically, we have unveiled a novel physical and functional interaction between Intu and STAT1. Specifically, Intu interacts with STAT1 likely at the basal body/centriole area

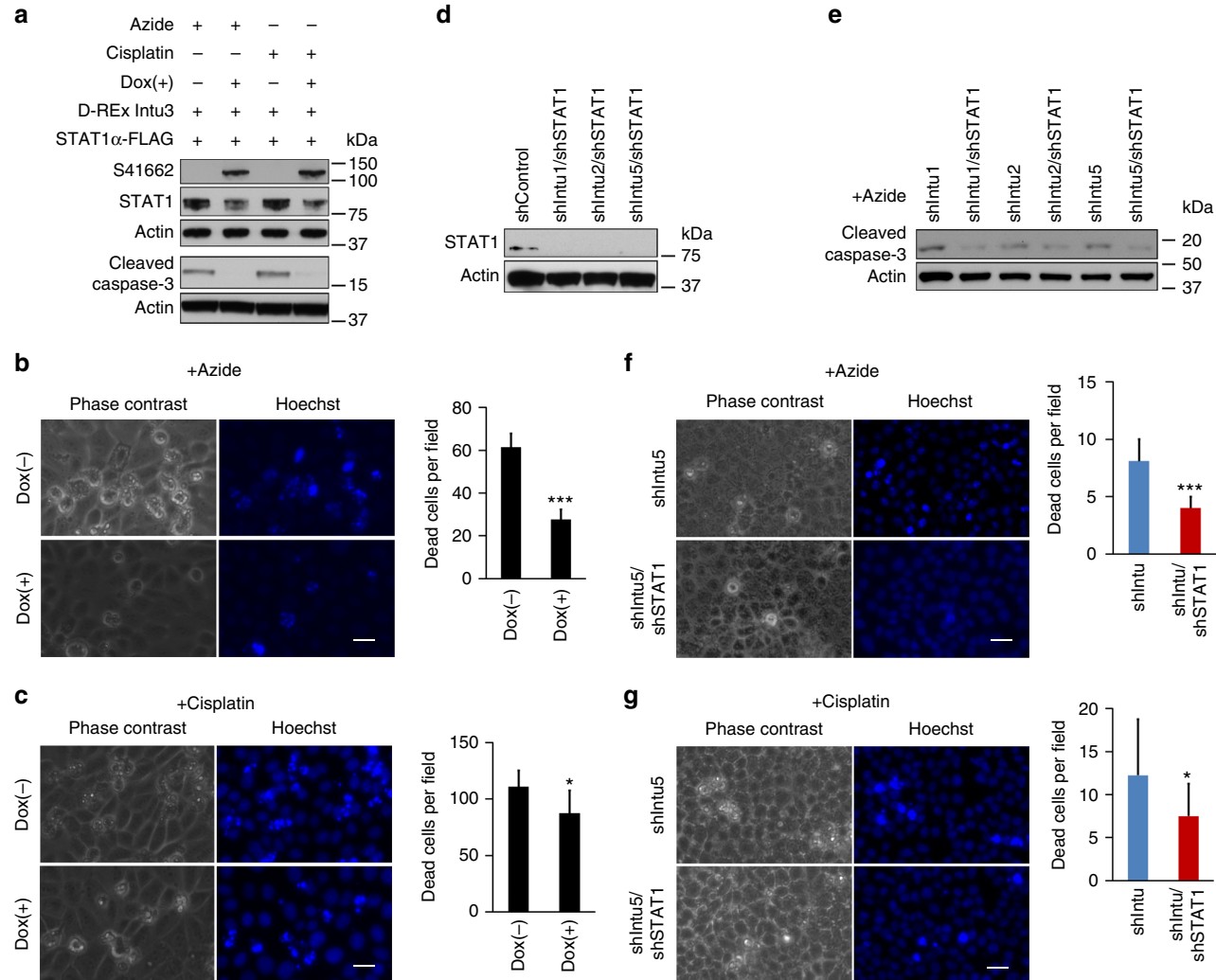

**Fig. 6** Intu protects cells from death through STAT1. **a** Inducible expression of Intu suppressed STAT1 expression and Caspase-3 cleavage during azide and cisplatin treatment. **b** Inducible expression of Intu suppressed azide-induced cell death in STAT1α-transfected cells ($n = 5$). **c** After exposure to cisplatin, Intu/STAT1 double overexpression cells showed resistance as compared to STAT1 single overexpression cells ($n = 8$). **d** Confirmation of STAT1 knockdown by shRNAs in Intu/STAT1 double knockdown cells. **e** Knockdown of STAT1 inhibited azide-induced Caspase-3 cleavage/activation in Intu-silenced cells. **f** Knockdown of STAT1 rescued Intu-silenced cells from azide-induced cell death ($n = 12$). **g** Knockdown of STAT1 rescued Intu-silenced cells from cisplatin-induced cell death ($n = 12$). Quantitative data are mean ± s.d. (error bars). Grouped $t$ test was used. $*p < 0.05$, $***p < 0.001$. Scale bars: 20 μm

to induce STAT1 degradation, ciliogenesis, and cell death resistance, whereas deficiency or blockade of Intu leads to ciliary defects, and STAT1 increase and accumulation in cell nucleus to promote cell death. As such, Intu protects cells and tissues under pathological conditions by inducing STAT1 degradation (Fig. 8).

The TAP assay suggested ~160 Intu-interacting proteins in BUMPT cells, among which ~45% of the proteins were also found by another group[11]. By delineating the mechanism of Intu-mediated protection, we confirmed STAT1 as a new interacting partner of Intu. STAT1 is a key member of the STAT family that consists of seven proteins (STAT1, 2, 3, 4, 5a, 5b, 6)[29–31]. STAT proteins are known to regulate apoptosis following phosphorylation by Janus kinase proteins[32]. STAT1 can be phosphorylated at both tyrosine 701 and serine 727 sites, and phosphorylated STAT1 forms homodimers or heterodimers with other STAT proteins, which translocate into the nucleus for gene transcription to regulate apoptosis[33]. Although tyrosine 701 phosphorylation is important in regulating gene transcription, serine 727 phosphorylation has been reported to maximize the apoptotic function of STAT1[32,34]. Our current study suggests that Intu can bind and

target STAT1 for degradation through the proteasome system, resulting in the decrease in both total and phosphorylated STAT1. Renal IRI is associated with the induction of Intu (Fig. 1), which may be an intrinsic protective or adaptive mechanism to antagonize the STAT1 pathway of apoptosis. Our results further support a critical role of serine 727 phosphorylation in STAT1 regulation (Fig. 5), which is consistent with the finding in heart IRI[34]. These observations are supported by the previous studies showing the role of STAT1 in kidney IRI[35–37].

As a CPLANE protein, Intu is known to regulate planar polarization in *Drosophila* and ciliogenesis[19,38]. However, ablation of *Intu* specifically from kidney proximal tubules does not reduce the number or length of primary cilia in mouse kidneys under control conditions (Fig. 7b). In contrast, primary cilia grow longer after kidney injury in WT mice, but not in KO mice (Fig. 7b). Thus, Intu may not be required for cilium maintenance, but it is important for ciliogenesis and ciliary elongation. Based on the current understanding, failure of primary cilia to elongate after renal IRI in KO mice may be caused by the lack of Intu-mediated IFT protein recruitment to the base of primary cilia[11].

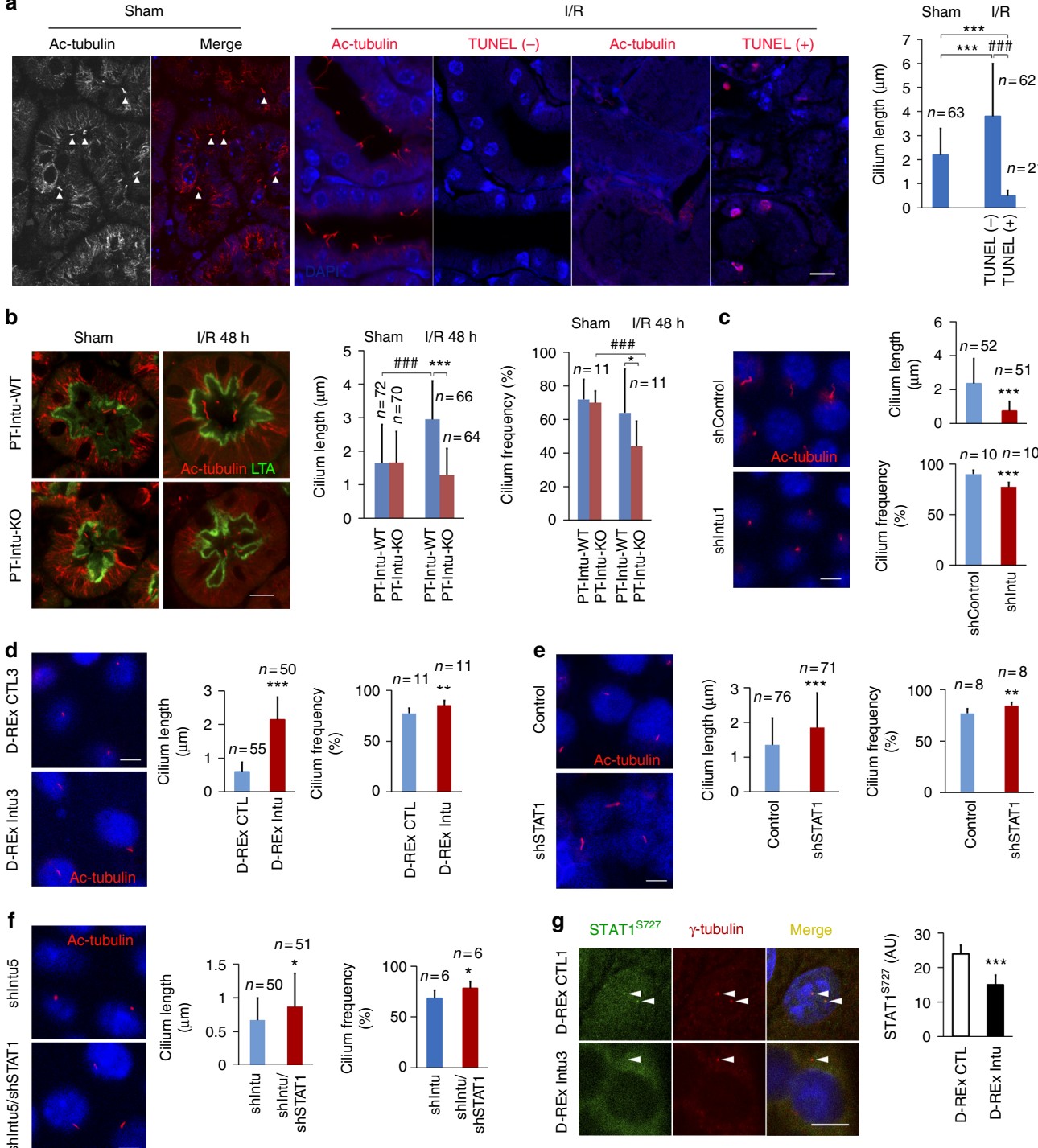

**Fig. 7** Regulation of primary cilia by Intu and STAT1. **a** Cilium length and cell death in kidney tubules during renal IRI in mice. WT mice were subjected to sham operation or 35 min of ischemia with 48 h of reperfusion to harvest kidneys for Ac-tubulin immunofluorescence staining and TUNEL assay. TUNEL-negative tubules in injured tissues had longer cilia compared to sham controls, while TUNEL-positive tubules had shorter cilia. White triangle denotes cilia. **b** Ac-tubulin and LTA staining showing cilium length and frequency in proximal tubules after renal IRI in WT mice that was attenuated in PT-Intu-KO mice. **c** Ac-tubulin staining showing shortened cilia and lower frequency in Intu-knockdown cells (shIntu), compared to control shRNA-transfected cells (shControl). **d** Intu overexpression (D-REx Intu3) preserved cilium length during azide treatment. **e** Longer cilia and higher cilium frequency in STAT1-knockdown cells as compared to control cells. **f** Intu/STAT1 double knockdown cells had longer cilium than Intu only knockdown cells. **g** Double immunofluorescence of serine 727 phosphorylated STAT1 and γ-tubulin (centriole marker) to show that inducible Intu expression abolished centriole-associated STAT1. Fluorescence signal was quantified with ImageJ ($n = 7$). $n$ is the number of cilia (**a–f**) or centrioles (**g**). Quantitative data are mean ± s.d. (error bars). Grouped $t$ test was used. $*p < 0.05$, $**p < 0.01$, $***p < 0.001$, $###p < 0.001$. Scale bar (µm): 10 (**a**, **b**, **g**), 5 (**c–f**)

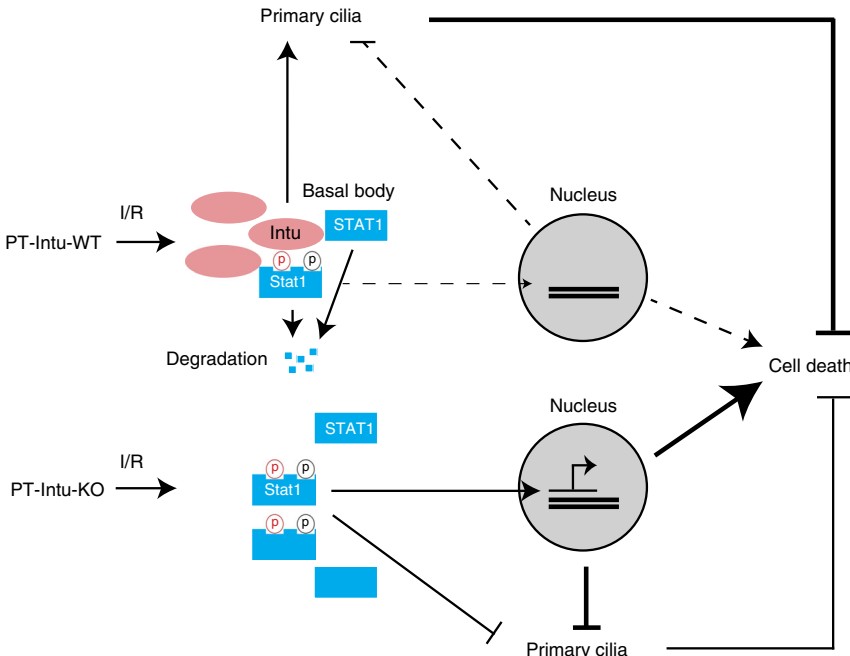

**Fig. 8** Schematic model of Intu/STAT1 regulation of cell death and cilium in kidney injury. In wild-type cells and tissues, renal IRI upregulates Intu, which binds and sequesters STAT1 in the basal body/centriole area for degradation, reducing phosphorylated STAT1 translocation to the nucleus and alleviating cell death (upper). Intu-KO cells and tissues are devoid of Intu, the STAT1 antagonizing mechanism, and as a result, more phosphorylated STAT1 accumulates in the nucleus to promote cell death (lower). Also, Intu promotes ciliogenesis, whereas STAT1 may have opposite effects. Phosphorylated STAT1 is marked with a circled P in red for Y701 and in black for S727

Our current study, however, suggests an alternate mechanism that involves Intu-STAT1 interaction. We show that STAT1 is normally present in centriole/basal body area and lost from this site after Intu overexpression. Moreover, knockdown of STAT1 increases cilium length and frequency (Fig. 7e). Therefore, it is possible that Intu may stimulate ciliogenesis by reducing STAT1 from centriole/basal body area. The mechanism whereby STAT1 regulates ciliogenesis remains elusive. Nonetheless, it may regulate ciliogenesis directly from the basal body through the Intu-IFT protein system and/or indirectly by transcribing relevant genes. Previous work reported the elongation of primary cilia during kidney repair or recovery after acute injury, but the underlying mechanism remains elusive[23–26]. Our results support a critical role of Intu in ciliogenesis and elongation following kidney injury. Another interesting observation in our study is that TUNEL-positive or death-associated cells tend to have shorter cilia and lower cilia frequency. This is particularly true for Intu-KO cells (Supplementary Fig. 10), suggesting that Intu may also play a role in preserving cilia during cell injury and death. Together, the results of this study suggest that Intu may promote cell survival and ciliogenesis in disease conditions by binding STAT1 to induce its degradation. Discovery of the Intu-STAT1 pathway offers new therapeutic targets. Future research may identify or develop specific pharmacological agents to modulate Intu-STAT1 interaction for the prevention and treatment diseases such as ischemic kidney injury.

## Methods

**Antibodies and reagents**. Intu antibody 1155PB4 (immunoblot (IB), 1:2500; IF, 1:100) was generated and characterized previously[27]. Intu antibody S4166 (IB, 1:1000–2500; IF, 1:100–500) was generated in this study by Proteintech. Briefly, a single KLH-conjugated mouse Intu peptide (DTDVEPEWLDSVQKNGEL) was used to immunize two prescreened rabbits. Harvested antisera from the rabbits were separately affinity-purified and designated as S41661 and S41662. Sources of other antibodies used in this study: antibodies to total STAT1 (14994; IB, 1:1000; IF, 1:100), phospho-STAT1 at serine 727 (STAT1$^{S727}$, 9177; IB, 1:1000; IF, 1:100),

Parp (9542; 1:1000), caspase-3 (9665; 1:1000), and cleaved caspase-3 (9661; 1:1000) from Cell Signaling Technology; c-Myc antibody (9E10; 1:1000) from Invitrogen; cyclophilin B antibody (CypB, ab16045; 1:5000) from Abcam; antibodies to FLAG (F3165; IB, 1:1000; IF, 1:500), acetylated tubulin (T7451; 1:1000), γ-tubulin (T5326; 1:500), and β-actin (actin, A2228; 1:10,000) from Sigma-Aldrich; anti-calmodulin-binding protein epitope tag antibody (CBP, 07-482; 1:1000) from Millipore; Bcl-2 (sc-783; 1:200) and Bcl-xL (sc-8392; 1:200) from Santa Cruz Technology; secondary antibodies for immunoblot (goat antimouse 31430, goat antirabbit 31460) from ThermoFisher Sci and for IF staining (goat antimouse 115-165-003, goat antirabbit 111-095-144) from Jackson ImmunoResearch. Fluorescein LTA (FL-1321) and PHA-E (FL-1121) were purchased from the Vector Labs. Prolong Gold antifade reagent with DAPI (4′-6-diamidino-2-phenylindole, P36931), doxycycline (D9891), MG132 (M7449), cisplatin (cis-diamineplatinum dichloride, 479306), azide (S2002), and carbonyl cyanide 3-chlorophenylhydrazone (C2759) were purchased from Sigma-Aldrich. Recombinant IFN-γ (PMC4031), blasticidin S HCl (R21001), zeocin (R25001), Hoechst 33342 (H1399), and Lipofectamine 2000 (12566014) were obtained from ThermoFisher Sci. Puromycin was purchased from Clontech. FuGENE 6 (E2693) was from Promega. NEMBUTAL (pentobarbital sodium injection, NDC 67386-501-55) was manufactured in Abbott Laboratories. Buprenex (buprenorphine hydrochloride, NDC 12496-0757-1) was produced in Reckitt Benckiser Pharmaceuticals Inc.

**Constructs and transfection**. The mouse Intu construct Intu-mycDDK was purchased from OriGene and subcloned into pNTAP vector to generate CBP-SBP-Intu construct for TAP assay. Intu was subcloned into pcDNA4/TO to establish doxycycline-inducible construct. PCR-based cloning was used to prepare deletion mutants of Intu, including CBP-SBP-IntuΔ181-942, CBP-SBP-IntuΔ268-942, and CBP-SBP-IntuΔ1-267. All STAT1 wild-type and mutation constructs, including STAT1α-FLAG, STAT1αY701F, STAT1αS727A, were obtained from Addgene[32]. shRNA constructs for Intu (shIntu) were purchased from Sigma-Aldrich. The constructs were transfected into cells with Lipofectamine 2000 or FuGENE 6.

**Proximal tubule-specific *Intu* knockout mice**. *Intu*-floxed (Intu$^{f/f}$) mice and PEPCK-Cre mice were generated previously[27,28]. Intu$^{f/f}$ mice were crossed with PEPCK-Cre mice to produce Intu$^{+/+}$/X$^{Cre}$X$^{Cre}$ and Intu$^{f/f}$/XY mice for further breeding (Fig. 1a). Intu$^{f/f}$/X$^{Cre}$Y mice were Intu-deficient in kidney proximal tubular cells. PT-Intu-KO mice and WT littermates (Intu$^{f/f}$/XY, PT-Intu-WT) (male, 8–12-week-old) were used for experiments. Mice were routinely housed in the animal facility of Charlie Norwood VA Medical Center at Augusta, GA. All animal experiments were performed according to a protocol approved by the Institutional Animal Care and Use Committee of Charlie Norwood VA Medical Center. Genomic DNA was extracted from mouse tail biopsy for PCR-based

genotyping. The *Intu*-floxed allele product was amplified as a DNA band of ~420 bp by PCR using three primers (5′-GATTAGGGTCTCGCCCTAGC-3′, 5′-ACAACCACAAGACTGCGTCA-3′, 5′-ATGCACAAGTGTGTGGGTGT-3′), while the WT product at ~200 bp. *Cre* gene product was detected at ~370 bp by PCR using one pair of primers (5′-ACCTGAAGATGTTCGCGATTATCT-3′, 5′-ACCGTCAGTACGTGAGATATCTT-3′).

**Renal ischemia-reperfusion**. Bilateral renal ischemia reperfusion was induced in mice[39,40]. Briefly, mice were anesthetized with pentobarbital (60 mg kg$^{-1}$) and kept on a homeothermic station to maintain body temperature at ~37 °C. Buprenorphine (0.05 mg kg$^{-1}$) was administered as analgesics for pain relief. Kidneys were surgically exposed for renal pedicle clamping with arterial clips to induce renal ischemia. After 35 min of clamping, the arterial clips were released for reperfusion. Sham control mice were subjected to identical procedure but without renal pedicle clamping. Blood was collected on first and second day after ischemia. During killing, kidneys were collected for histological and biochemical analysis.

**Renal function and histology**. Renal function was indicated by the levels of BUN and SCr. Briefly, blood samples were collected, kept at room temperature for clotting, and then centrifuged at 12,000×*g* for 5 min to obtain serum. BUN and SCr in serum samples were measured with the commercial kits from Stanbio Laboratories. The histology of kidney tissues was examined by hematoxylin and eosin (H&E) staining. Briefly, collected kidney tissues were fixed in 4% paraformaldehyde at 4 °C for 24 h and stored in 70% ethanol. The tissues were paraffin-embedded and sectioned at 4 μm thickness for H&E staining. Histologic changes were evaluated by the percentage of renal tubules showing tubular lysis, loss of brush border, cast formation, and other signs of injury.

**Cell culture, treatments, and morphological analysis of cell death**. BUMPT (originally obtained from Drs. Lieberthal and Shwartz at Boston University) and 293FT (Invitrogen) cells were cultured in DMEM with 10% FBS media. RPTC cells (originally obtained from Dr. Hopfer at Case Western Reserve University) were maintained in a specialized medium[41]. For cisplatin treatment, cells were incubated with 20 μM cisplatin. For azide or CCCP treatment, cells were incubated with 20 mM azide or 20 μM CCCP in glucose-free buffer for 3 h and then returned to full cell culture medium for 3 h of "reperfusion." For IFN-γ treatment, cells were exposed to 5 ng ml$^{-1}$ IFN-γ in culture media for 30 min. For morphological analysis of cell death, cells were stained with Hoechst 33342 for 2 min and then examined microscopically with the EVOS cell image system (ThermoFisher Sci). Cellular and nuclear morphologies were recorded by phase contrast and fluorescence microscopy. Typical cell death was indicated by cellular shrinkage and blebbing, and nuclear condensation and fragmentation.

**Tandem affinity purification and mass spectrometry**. TAP was done with the InterPlay Mammalian TAP System from Agilent Technologies. Briefly, BUMPT cells were transfected with CBP-SBP-Intu or empty vector. After 24 h, the cells were trypsinized, centrifuged, and resuspended in lysis buffer for protein extraction. STV resin was then added into the cell lysate for 2 h of incubation at 4 °C. Subsequently, STV resin was collected, washed completely, and resuspended in SEB buffer for 30 min to elute the protein complexes. Elutes were incubated with calmodulin resin at 4 °C for 2 h to allow the protein complexes to bind. Finally, the binding proteins were eluted by boiling for 5 min after addition of the loading buffer and subjected to 4–12% SDS-PAGE to resolve the interacting proteins. The gel was silver-stained with the SilverQuest Silver Staining Kit from Life Technologies. The protein bands detected in CBP-SBP-Intu-transfected cells, but not in empty vector transfected cells, were extracted. Proteins were identified by microcapillary LC-MS/MS in the Taplin Mass Spectrometry Facility (Harvard Medical School).

**Streptavidin-pulldown, co-IP, and immunoblot analysis**. For STV-pulldown assay, 40 μl STV resin slurry was washed to remove the ethanol storage buffer, incubated with the protein samples for 2 h at 4 °C, and then collected by centrifugation at 1500×*g* for 5 min. For co-IP, one specific antibody and agarose beads were added to cell lysate for 2 h of incubation at 4 °C under constant mixing and then collected by centrifugation. Following extensive washes, the agarose resin was boiled in sample loading buffer with SDS to release the bound proteins. The protein samples were subjected to SDS-PAGE and transferred to Immun-Blot PVDF membrane (Bio-Rad). The blot was incubated with the blocking buffer containing 5% non-fat milk or 3% bovine serum albumin (BSA), and then primary antibody. After washing with phosphate buffer saline (PBS), the blot was incubated with HRP-conjugated secondary antibody at room temperature. After further washing, proteins on the blot were detected with the SuperSignal West Pico Chemiluminescent Substrate (ThermoFisher Sci) or Clarity Western ECL Substrate (Bio-Rad). Uncropped images of the most important blots in this study are included in Supplementary Fig. 11.

**Immunofluorescence, TUNEL, and confocal microscopy**. For tissue section processing, kidney sections were dried overnight before deparaffinization and rehydration through graded concentrations of ethanol. Antigen retrieval was done by boiling slides in the retrieval buffer (10 mM sodium citrate pH 6.0, 0.05% Tween-20) for 1 h. For cultured cell fixation, cells were fixed in 4% paraformaldehyde/2% sucrose solution and permeabilized with 0.1% Triton X-100 for 5 min. Alternatively, samples were fixed with methanol at −20 °C for 20 min as before[25,42]. Cells and tissue sections were blocked with BSA for 1 h. After incubation with primary antibodies for 1–2 h, the sections were washed with PBS, and then incubated with secondary antibodies for 1–1.5 h. Slides were mounted with the Prolong Gold antifade reagent containing DAPI or Hoechst 33342 for nuclear staining. TUNEL assay was done with the In Situ Cell Death Detection Kit from Roche. For image analysis, Zeiss Axio fluorescence and confocal microscopes (Carl Zeiss Inc.) were used. The confocal microscope was equipped with the LSM Image analysis system that was used for the measurement of the cilium length and quantification of cilium frequency. Cilium length was measured on maximum intensity projection images obtained from 4 μm kidney sections. Measurements did not account for potential curvature in the Z-axis. Briefly, cilium length was shown in collected images by drawing a line along the primary cilia. If cilium length was too short, images would be expanded so that the length was measurable. To maximize the accuracy, >50 cilia were measured in each condition.

**Doxycycline-inducible expression of Intu in RPTC cells**. To establish Intu-inducible cell lines, RPTC cells were first transfected with pcDNA6 and incubated with blasticidin (10 μg ml$^{-1}$) to select stably transfected cell clones. A total of 14 clones were selected and tested by immunostaining after pcDNA4-Intu transient transfection with or without doxycycline addition. Five clones were further tested by immunoblotting to determine which clone had tightly controlled Intu expression. Clone 4 was used for further establishment of doxycycline-inducible cells. After selection with zeocin (1 mg ml$^{-1}$), we verified nine clones by immunoblotting and immunostaining.

**Single- and double-gene Knockdown BUMPT cells**. For STAT1 knockdown in BUMPT cells, we purchased validated STAT1 transduction particles from Sigma-Aldrich. Intu-knockdown transduction particles were generated based on the standard protocol. For double Intu/STAT1 knockdown, we sequentially transduced BUMPT cells with both types of knockdown transduction particles. Cells were selected and maintained with 2.5 μg ml$^{-1}$ and 1 μg ml$^{-1}$ puromycin, respectively.

**RNA isolation, reverse transcription, and quantitative PCR**. Total RNA was isolated from kidney cortex tissues or cultured BUMPT cells according to the instructions in mirVana miRNA Isolation Kit (Life Technologies, Carlsbad, CA). Reverse transcription into cDNA was done with 1 μg RNA by using iScript cDNA Synthesis Kit (Bio-Rad). qRT PCR was performed by a StepOne Real-time System (Applied Biosystems, Foster City, CA) with iTaq Universal SYBR Green Supermix (Bio-Rad). Each gene reaction was performed in triplicate and data were normalized by the internal control β-actin. The genes and primers used in the experiments are, *Bcl-2* (5′-GGAGGATTGTGGCCTTCTTT-3′, 5′-GAGACAGCC AGGAGAAATCAA-3′), *Caspase-1* (5′-ACAAGATCCTGAGGGCAAAG-3′, 5′-CAGCTGATGGAGCTGATTGA-3′), *Intu* (5′-GTACAAGGATGTGACCGT CTA-3′, 5′-GATGATCCCCACCAGTACTTC-3′), and *β-actin* (5′-GACTCAT CGTACTCCTGCTTG-3′, 5′-GATTACTGCTCTGGCTCCTAG-3′).

**Statistics**. Data were expressed as mean ± s.d. Paired or grouped *t* test was calculated. *p* < 0.05 was considered statistically significant.

**Data availability**. All relevant data of this study are available from the authors.

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

# ARTICLE

7. Seifert, J. R. & Mlodzik, M. Frizzled/PCP signalling: a conserved mechanism regulating cell polarity and directed motility. *Nat. Rev. Genet.* **8**, 126–138 (2007).

8. Nauli, S. M. et al. Polycystins 1 and 2 mediate mechanosensation in the primary cilium of kidney cells. *Nat. Genet.* **33**, 129–137 (2003).

9. Wallingford, J. B. & Mitchell, B. Strange as it may seem: the many links between Wnt signaling, planar cell polarity, and cilia. *Genes Dev.* **25**, 201–213 (2011).

10. Yasunaga, T. et al. The polarity protein Inturned links NPHP4 to Daam1 to control the subapical actin network in multiciliated cells. *J. Cell Biol.* **211**, 963–973 (2015).

11. Toriyama M. et al. The ciliopathy-associated CPLANE proteins direct basal body recruitment of intraflagellar transport machinery. *Nat. Genet.* **48**, 648–656 (2016).

12. Zilber, Y. et al. The PCP effector fuzzy controls cilial assembly and signaling by recruiting Rab8 and dishevelled to the primary cilium. *Mol. Biol. Cell* **24**, 555–565 (2013).

13. Cui, C. et al Wdpcp, a PCP protein required for ciliogenesis, regulates directional cell migration and cell polarity by direct modulation of the actin cytoskeleton. *PLoS Biol.* **11**, e1001720 (2013).

14. Saburi, S. et al. Loss of Fat4 disrupts PCP signaling and oriented cell division and leads to cystic kidney disease. *Nat. Genet.* **40**, 1010–1015 (2008).

15. Badano, J. L., Mitsuma, N., Beales, P. L. & Katsanis, N. The ciliopathies: an emerging class of human genetic disorders. *Annu. Rev. Genomics Hum. Genet.* **7**, 125–148 (2006).

16. Hildebrandt, F., Benzing, T. & Katsanis, N. Ciliopathies. *N. Engl. J. Med.* **364**, 1533–1543 (2011).

17. Guirao, B. et al. Coupling between hydrodynamic forces and planar cell polarity orients mammalian motile cilia. *Nat. Cell Biol.* **12**, 341–350 (2010).

18. Tissir, F. et al. Lack of cadherins Celsr2 and Celsr3 impairs ependymal ciliogenesis, leading to fatal hydrocephalus. *Nat. Neurosci.* **13**, 700–707 (2010).

19. Park, T. J., Haigo, S. L. & Wallingford, J. B. Ciliogenesis defects in embryos lacking inturned or fuzzy function are associated with failure of planar cell polarity and Hedgehog signaling. *Nat. Genet.* **38**, 303–311 (2006).

20. Menezes, L. F. & Germino, G. G. Polycystic kidney disease, cilia, and planar polarity. *Methods Cell Biol.* **94**, 273–297 (2009).

21. Patel, V. et al. Acute kidney injury and aberrant planar cell polarity induce cyst formation in mice lacking renal cilia. *Hum. Mol. Genet.* **17**, 1578–1590 (2008).

22. Takakura, A. et al. Renal injury is a third hit promoting rapid development of adult polycystic kidney disease. *Hum. Mol. Genet.* **18**, 2523–2531 (2009).

23. Verghese, E., Weidenfeld, R., Bertram, J. F., Ricardo, S. D. & Deane, J. A. Renal cilia display length alterations following tubular injury and are present early in epithelial repair. *Nephrol. Dial. Transplant.* **23**, 834–841 (2008).

24. Verghese, E. et al. Renal primary cilia lengthen after acute tubular necrosis. *J. Am. Soc. Nephrol.* **20**, 2147–2153 (2009).

25. Wang, S., Wei, Q., Dong, G. & Dong, Z. ERK-mediated suppression of cilia in cisplatin-induced tubular cell apoptosis and acute kidney injury. *Biochim. Biophys. Acta* **1832**, 1582–1590 (2013).

26. Wang, S. & Dong, Z. Primary cilia and kidney injury: current research status and future perspectives. *Am. J. Physiol. Ren. Physiol.* **305**, F1085–F1098 (2013).

27. Zeng, H., Hoover, A. N. & Liu, A. PCP effector gene Inturned is an important regulator of cilia formation and embryonic development in mammals. *Dev. Biol.* **339**, 418–428 (2010).

28. Rankin, E. B., Tomaszewski, J. E. & Haase, V. H. Renal cyst development in mice with conditional inactivation of the von Hippel-Lindau tumor suppressor. *Cancer Res.* **66**, 2576–2583 (2006).

29. Hou, S. X., Zheng, Z., Chen, X. & Perrimon, N. The Jak/STAT pathway in model organisms: emerging roles in cell movement. *Dev. Cell* **3**, 765–778 (2002).

30. Kisseleva, T., Bhattacharya, S., Braunstein, J. & Schindler, C. W. Signaling through the JAK/STAT pathway, recent advances and future challenges. *Gene* **285**, 1–24 (2002).

31. O'Shea, J. J., Gadina, M. & Schreiber, R. D. Cytokine signaling in 2002: new surprises in the Jak/Stat pathway. *Cell* **109**, S121–S131 (2002).

32. Wen, Z., Zhong, Z. & Darnell, J. E. Jr. Maximal activation of transcription by Stat1 and Stat3 requires both tyrosine and serine phosphorylation. *Cell* **82**, 241–250 (1995).

33. Kim, H. S. & Lee, M. S. STAT1 as a key modulator of cell death. *Cell Signal.* **19**, 454–465 (2007).

34. Stephanou, A. et al. Induction of apoptosis and Fas receptor/Fas ligand expression by ischemia/reperfusion in cardiac myocytes requires serine 727 of the STAT-1 transcription factor but not tyrosine 701. *J. Biol. Chem.* **276**, 28340–28347 (2001).

35. Si, Y. et al. Dexmedetomidine protects against renal ischemia and reperfusion injury by inhibiting the JAK/STAT signaling activation. *J. Transl. Med.* **11**, 141 (2013).

36. Yang, N. et al. Blockage of JAK/STAT signalling attenuates renal ischaemia-reperfusion injury in rat. *Nephrol. Dial. Transplant.* **23**, 91–100 (2008).

37. Tang, S. C. et al. Renoprotection by rosiglitazone in accelerated type 2 diabetic nephropathy: role of STAT1 inhibition and nephrin restoration. *Am. J. Nephrol.* **32**, 145–155 (2010).

38. Lee, H. & Adler, P. N. The function of the frizzled pathway in the *Drosophila* wing is dependent on inturned and fuzzy. *Genetics* **160**, 1535–1547 (2002).

39. Zhang, D. et al. Tubular p53 regulates multiple genes to mediate AKI. *J. Am. Soc. Nephrol.* **25**, 2278–2289 (2014).

40. Wei, Q. et al. Targeted deletion of dicer from proximal tubules protects against renal ischemia-reperfusion injury. *J. Am. Soc. Nephrol.* **21**, 756–761 (2010).

41. Brooks, C., Wei, Q., Cho, S. G. & Dong, Z. Regulation of mitochondrial dynamics in acute kidney injury in cell culture and rodent models. *J. Clin. Invest.* **119**, 1275–1285 (2009).

42. Wang, S., Livingston, M. J., Su, Y. & Dong, Z. Reciprocal regulation of cilia and autophagy via the MTOR and proteasome pathways. *Autophagy* **11**, 607–616 (2015).

## Acknowledgements

We thank Dr. Haase at Vanderbilt University School of Medicine (Nashville, TN) for providing the PEPCK-Cre mouse line, Drs. Lieberthal and Shwartz at Boston University for providing the BUMPT cell line, Dr. Hopfer at Case Western Reserve University for providing the RPTC cell line. This study was supported in part by grants from the National Natural Science Foundation of China (81430017), the National Institutes of Health (DK058831, DK087843) of USA, and the Department of Veterans Administration of USA (BX000319), Carlos and Marguerite Mason Trust.

## Author contributions

S.W. and Z.D. designed the experiments. S.W. performed the experiments. S.W., G.W., H.-F.D., S.H., S.N., and Z.D. analyzed the results. S.W., A.L., G.W., H.-F.D., S.H., S.N., and Z.D. wrote the manuscript.

## Additional information

**Competing interests:** The authors declare no competing interests.

