## [Peer Review File(PDF 665 kb) · Nature Communications]

Reviewers' comments:

Reviewer #1 (Remarks to the Author):

The manuscript of Wang et al. studies the impact of the ciliary and PCP protein Intu primarily in proximal tubule cells of the kidney. They suggest a model, in which Intu might protect cells from ischemia-reperfusion damage while its interactor Stat1 might do the opposite. Intu has been previously implicated with human ciliopathies in a study that described the so called CPLANE genetic module (ciliogenesis and planar polarity effector). The main components of this module have been described to be interturned (Intu), Fuzzy, and WDPCP.

This is an interesting story on an interesting topic. However, I have some major concerns with the presented data:

Major points:

- 1) The authors suggest Stat1 to be a critical player in the I/R damage modulated by Intu: Is there any change in expression of Stat1 transcriptional targets in WT vs. Intu KO mice? Is the transcriptional activity of Stat1 really altered *in vivo*? And does Intu affect phosphorylation status of Stat1?
- 2) The data presented in Figure 1 describing the mouse model is interesting but not convincing. The authors should demonstrate that the knockout in prox. tubule cells is efficient. To this end, the n=1 western blot presented in Fig. 1c is not sufficient. More animals and a proper quantification should be added (with convincing loading controls); and in addition, the knockout should be proven, e.g. in isolated prox. tubule cells or in IHC/IF.
- 3) Fig. 2 suggests a role of Intu in apoptosis, which is not convincingly shown. The LTA staining in Fig. 2a is not really clear. The pattern does not look like prox. tubules. The authors should provide better pictures (lower magnifications; these pictures should be as clear as in Fig. 7b) and add additional markers. Ultimately, crossing these mice with a reporter strain (e.g. mG/mT) would help to identify cre positive KO cells *in vivo*. This would make a strong point. In addition, the validation of the shRNA in Fig. 2b is not convincing, nor is the effect presented in 2d. This experiment should be performed with at least two independent, validated shRNAs and the authors should try to rescue the effect by re-expression of INTU. Moreover, the conclusion at this point can't be "apoptosis", but should be "cell death". Additional data and experiments are needed to allow for distinguishing between diff. types of regulated cell death (apoptosis, necroptosis, ferroptosis, etc) and simple passive cell death / necrosis. There are very similar issues with Fig. 6.
- 4) Interactome data (Fig 4): What are the additional proteins pulled down with TAP.Intu? In the initial study on the CPLANE module (Nat Gen 2016) an entire interaction network has been described. Does the TAP screen confirm some of these interactions? In fact, Stat1 has been pulled down with Fuz and Tulp4 in this previous study suggesting that it is a component of the complex. Is there any evidence beyond the mapping to the PDZ domain for a direct interaction of Intu and Stat1? In addition, the increase in Azide and Cisplatin treated cells is hardly visible in the figure.
- 5) The localization of Intu at the centrosome / basal body (BB) has been demonstrated before (Toriyama et al. , Nat Gen 2016), although not in prox. tubule cells. However, the authors show Intu at the centrosome / BB, and Stat1 at the same localization. This does not prove that their interaction really takes place at this compartment. This conclusion must be drawn more carefully, or additional data has to be added proving the interaction (e.g. Duolink Proximity-Ligation-Assays).
- 6) Why does Intu not affect the Stat1 expression levels in Fig 4b? This is in contrast to Fig. 5a...
- 7) The IF pictures in Fig. 7a do not allow to identify cilia nor to analyze ciliary length. Better

quality must be provided. How was the quantification done? And talking about ciliogenesis: to what extent does KO or OE of Intu and/or Stat1 affect the number of ciliated cells?

General comments:

- In many western blots molecular weight standards are missing making it difficult to draw any conclusion out of the data (e.g. Fig 4b, Fig 3, ...)
- Several western blots lack controls and the quantification does not always fit to the presented blot. E.g. the quantification of Parp/actin in Fig. 5e. This is not convincing. In any case, solid densitometric analyses might be nearly impossible in case bands are not clearly separated.
- The manuscript requires some language editing (e.g. "Kidney tissues are characterized by primary cilia and PCP... " (pg 3) etc.
- "the site of ciliogenesis" is not the very best description for the basal body.
- The authors should more clearly distinguish between "ciliogenesis" and "ciliary elongation". If Intu is involved in cilia elongation ( longer cilia) , that would not necessarily mean it has a function in ciliogenesis, which typically refers to the very early step, the "birth" of this lovely organelle.

Reviewer #2 (Remarks to the Author):

This interesting manuscript proposes a new twist to a pathway that protects renal tubule cells from apoptosis during recovery from ischemia/reperfusion injury. The authors find that the CPLANE protein Intu, a component of the basal body previously proposed to load Ift proteins into cilia, regulates STAT1. After injury, STAT1 is activated to promote the apoptotic injury response. The authors suggest that Intu protects from this response by binding and promoting the proteasome dependent degradation of STAT1. In addition, tubule cells that recover from injury elongate their cilia, while those that do not display shorter cilia. While Intu is associated with ciliogenesis in other contexts, its absence in tubules has no effect on wildtype cilia length. However, Intu protects cilium length after injury, and here again this function is antagonistic with STAT1.

The story is important, and in principle I'm in favor of publication. However, There are several critical documentation issues that must first be addressed. In addition, several questions should be answered, and some issues with the introduction should be addressed.

First, the documentation problems:

Definitions of precisely what is being quantified is missing for most graphs. For example, in Fig 2a, the number of apoptotic cells – what is the denominator? Similar definitions are needed for most of the subsequent graphs as well.

The confocal images are of marginal quality, and I'm struggling to see how the authors can reliably quantify cilium length, at least from images like the ones provided. Cilia tend to bend and point down the long axis of the tubule, and thus their length is very difficult to assess except in longitudinal slices. Some description of how the measurements were made, along with high quality images, is needed.

Along those lines, the authors show that cilium response is opposite in apoptotic and non-apoptotic cells. However, in subsequent experiments, the data are not divided between apoptotic and non-apoptotic cells. This seems to be an important distinction that may segregate the data in a meaningful way, and should be addressed.

In addition to the above documentation, the following questions should be answered:

Line 77: Intu was induced in WT kidneys but not in KO kidneys. Is there no expression in more distal nephron segments? Since the KO is specific to proximal tubules, one might expect to see some induction from the unaffected segments.

How do the authors explain that STAT1 is the major band on the Fig 4a TAP pulldown gel? What about the rest of the CPLANE complex? This is quite unexpected.

Do the authors agree that the subsequent pull down assays suggest a weak interaction with only a small fraction bound?

How do the authors explain the apparent cloud of STAT1 immunofluorescence near but not bound to the basal bodies?

The introduction:

Line 46: "The PCP core proteins include those of the Frizzled and the Fat systems 1, 7." In the PCP literature, the "core system" refers to proteins of the Fz, Vangl, Celsr, etc group, while the Fat system is not considered part of the core but a separate mechanism.

Line 48 "Recently, PCP has been linked to primary cilia." This is a complex and controversial area, and in general the idea that there is an intimate association is suspect. The authors should look at reviews by John Wallingford, and provide a more nuanced discussion.

Line 59 "Kidney tissues are characterized by primary cilia and PCP, and dysregulation of cilia and PCP contributes to the pathogenesis of renal diseases, such as polycystic kidney disease 19." Similarly, this idea is complex, and while embedded in the literature, there is little direct evidence. Again, a more circumspect telling is warranted.

Finally, the authors fail to acknowledge prior work indicating that STAT1 is involved in the injury response. For example, Yang et al 2008 Jan;23(1):91-100, Si et al J Transl Med. 2013 Jun 9;11:141, and perhaps Tang et al AJN 2010;32(2):145-55 should be cited.

Reviewer #3 (Remarks to the Author):

This study analyzes the role of the CPLANE protein, Intu, in recovery from renal ischemia-reperfusion injury. Intu is thought to have a role in ciliogenesis and planar cell polarity (PCP) and has been implicated in ciliopathy disorders in frog and humans. Here, Intu is ablated in the proximal tubule (PT), aggravating recovery from renal ischemia-reperfusion injury due to defective ciliogenesis. Interaction of Intu with STAT1 at the base of the cilium is proposed to regulate STAT1 degradation. Hence, a process of recovery via targeting STAT1 for degradation is proposed.

Specific Points:

1. At what age does the PEPCK-Cre inactivate Intu in the PT? Is it after completion of kidney development? Are any developmental defects found in these kidneys? Do these animals have PCP defects?
2. Since knockout mice are available, it seem strange that shRNA, with incomplete silencing, was used for the cellular analysis in Figure 2.
3. Further data about the TAP study should be shown. Were other interacting proteins identified? How do they compare to the proteins identified by Toriyama et al?
4. Interaction by co-IP between endogenous Intu and STAT1 should be shown. In addition, the role of Intu in targeting STAT1 for degradation should be analyzing by loss of the endogenous protein.

5. The cilium length differences in Figure 7 seem marginal, and measuring cilia length differences in cultured cells is fairly unreliable. Data from STAT1 knockout mice would strengthen this analysis.
6. The study would be more complete if the information found from the Intu/STAT1 interaction could be translated back into a treatment option that is shown to be of value for treatment of ischemia-reperfusion injury in vivo.

Point-by-point Response to Reviewers' Comments

We thank the reviewers for recognizing the novelty and potential significance of this study. We also appreciate the constructive criticisms and suggestions, which have guided this revision to significantly improve the manuscript. The revisions are highlighted in red font. Here lists our point-by-point response.

Reviewer 1

“The manuscript of Wang et al. studies the impact of the ciliary and PCP protein Intu primarily in proximal tubule cells of the kidney. They suggest a model, in which Intu might protect cells from ischemia-reperfusion damage while its interactor Stat1 might do the opposite. Intu has been previously implicated with human ciliopathies in a study that described the so called CPLANE genetic module (ciliogenesis and planar polarity effector). The main components of this module have been described to be intuned (Intu), Fuzzy, and WDPCP.

This is an interesting story on an interesting topic. However, I have some major concerns with the presented data”

Response: We thank the reviewer for appreciating the main findings in our study. We have addressed the concerns as summarized below.

“1) The authors suggest Stat1 to be a critical player in the I/R damage modulated by Intu: Is there any change in expression of Stat1 transcriptional targets in WT vs. Intu KO mice? Is the transcriptional activity of Stat1 really altered in vivo? And does Intu affect phosphorylation status of Stat1?”

Response: To address this, we first compared WT and Intu-KO mice kidneys for the expression of Bcl-2 and Bcl-XL, two transcriptional repression targets of STAT1 (Stephanou A. et al. Cell Death Differentiation 7:329-10, 2000). Compared to WT, Intu-KO kidney tissues showed lower Bcl-2 and Bcl-XL expression (Fig. 5e), which is consistent with STAT1 activation and consequent repression of Bcl-2/Bcl-XL expression in Intu-KO tissues. By quantitative PCR, we further showed that Intu-KO kidney tissues expressed less Bcl-2 mRNA, but significantly more caspase-1 (a transcriptional activation targets of STAT1: Chin Y. et al. Mol Cell Biol 17:5328-37, 1997) (Suppl Fig. 7). Together, these results provide further evidence of STAT1 activation in Intu-KO kidney tissues in vivo. In addition, we verified the increase of STAT1 phosphorylation (especially at S727) in Intu-KO kidneys (Fig. 5e).

“2) The data presented in Figure 1 describing the mouse model is interesting but not convincing. The authors should demonstrate that the knockout in prox. tubule cells is efficient. To this end, the n=1 western blot presented in Fig. 1c is not sufficient. More animals and a proper quantification should be added (with convincing loading controls); and in addition, the knockout should be proven, e.g. in isolated prox. tubule cells or in IHC/IF.”

Response: As suggested, we have now presented 3 pairs of WT and PT-Intu-KO mice kidney tissues; we have also quantified Intu signal by densitometry to verify >80% KO efficacy in this model (Fig. 1c). Intu knockout in proximal tubules has been further confirmed by immunofluorescence staining (Suppl Fig. 2). Of note, the PEPCK-Cre model used in our study induces gene ablation in the majority (~80%) of kidney proximal tubule cells, but not in other kidney cells (Rankin EB, et al. Cancer Res 66, 2576-2583, 2006). By using this mouse line, we have successfully established several kidney proximal tubule-specific gene knockout models (Wei Q. JASN 2010; Jiang M. Kidney Int 2012; Zhang D. JASN 2014. Livingston M. Autophagy 2016).

“3) Fig. 2 suggests a role of Intu in apoptosis, which is not convincingly shown. The LTA staining in Fig. 2a is not really clear. The pattern does not look like prox. tubules. The authors should provide better pictures (lower magnifications; these pictures should be as clear as in Fig. 7b) and add additional markers. Ultimately, crossing these mice with a reporter strain (e.g. mG/mT) would help to identify cre positive KO cells in vivo. This would make a strong point. In addition, the validation of the shRNA in Fig. 2b is not convincing, nor is the effect presented in 2d. This experiment should be performed with at least two independent, validated shRNAs and the authors should try to rescue the effect by re-expression of INTU. Moreover, the conclusion at this point can't be “apoptosis”, but should be “cell death”. Additional data and experiments are needed to allow for distinguishing between diff. types of regulated cell death (apoptosis, necroptosis, ferroptosis, etc) and simple passive cell death / necrosis. There are very similar issues with Fig. 6.”

Response:

We have replaced Fig. 2a with new images of better quality. We have also added staining images at a lower magnification (Suppl Fig. 3). LTA is widely used as a marker for kidney proximal tubules. Like other proximal tubule markers, LTA staining pattern is often disrupted during kidney tubular injury, and consequently the staining is somewhat unclear and distorted. We have tested another proximal tubule marker PHA-E (Phaseolus vulgaris erythroagglutinin, Suppl Fig. 3).

We agree it would make a stronger point to use mG/mT mice to further verify the association of cell death and cilia in the Cre-positive/KO cells. But we and others have confirmed the high gene targeting specificity and efficacy of the PEPCK-Cre model in multiple studies (Rankin EB. Cancer Res 2006; Wei Q. JASN 2010; Jiang M. Kidney Int 2012; Zhang D. JASN 2014. Livingston M. Autophagy 2016). In addition, the mG/mT mouse line is not currently available to us. We contacted the Jackson Lab and was informed that they need several months to cryo-recover the mG/mT mice, which will then need to breed with our PT-Into-KO mice; by estimation it would take at least a year to generate the model for renal ischemia study. Thus, we hope to have the reviewer's approval not to require the complementary, yet time-consuming, mG/mT test in the current study.

We have further confirmed Intu knockdown after stably transfection of 5 different shRNA sequence plasmids by qRT-PCR (Fig. 2b). We have also presented the effects of 3 knockdown clones (Fig. 2d, Fig. 2e). We considered the idea of re-expressing Intu in knockdown cells, but it is apparently very challenging because the exogenous Intu is also the knockdown target of stably transfected Intu shRNAs. As suggested, we have changed ‘apoptosis’ to ‘cell death’ throughout the manuscript.

“4) Interactome data (Fig 4): What are the additional proteins pulled down with TAP.Intu? In the initial study on the CPLANE module (Nat Gen 2016) an entire interaction network has been described. Does the TAP screen confirm some of these interactions? In fact, Stat1 has been pulled down with Fuz and Tulp4 in this previous study suggesting that it is a component of the complex. Is there any evidence beyond the mapping to the PDZ domain for a direct interaction of Intu and Stat1? In addition, the increase in Azide and Cisplatin treated cells is hardly visible in the figure.”

Response: Yes, many other proteins were pulled down in our TAP assay. We have now provided the representative image of a full-lane silver staining gel to show the major bands (Suppl Fig. 4). We have also provided a list of the proteins identified by Mass Spec (Suppl Table 1). Actually, we revealed many identical CPLANE

interactome proteins initially reported by Wallingford and colleagues in their 2016 Nature Genetics paper. We currently do not have evidence beyond the mapping to the PDZ domain for a direct interaction between Intu and STAT1. We have replaced the immunoblot in Fig. 4f to show Intu increase in azide- and cisplatin-treated cells.

“5) The localization of Intu at the centrosome / basal body (BB) has been demonstrated before (Toriyama et al. , Nat Gen 2016), although not in prox. tubule cells. However, the authors show Intu at the centrosome / BB, and Stat1 at the same localization. This does not prove that their interaction really takes place at this compartment. This conclusion must be drawn more carefully, or additional data has to be added proving the interaction (e.g. Duolink Proximity-Ligation-Assays).”

Response: We thank the reviewer for this thoughtful comment and, accordingly, we have revised the description not to make the claim.

6) Why does Intu not affect the Stat1 expression levels in Fig 4b? This is in contrast to Fig. 5a....

Response: This was caused by the difference in the experimental conditions. In Fig 4b experiment, CBP-SBP-Intu was transiently transfected into BUMPT cells. Because the transfection efficacy in these cells was about 15%, STAT1 in the majority (85%) of the cells in the dish was not affected. In contrast, in Fig. 5a experiment, Intu-mycDDK and STAT1-FLAG were co-transfected into BUMPT cells. Although only ~15% cell got transfected, Intu-mycDDK and STAT1-FLAG were transfected into the same subset of cells (the cells susceptible to transfection). In these co-transfected cells, Intu-mycDDK are expected to suppress STAT1-FLAG expression, which is shown in Fig 5b.

“7) The IF pictures in Fig. 7a do not allow to identify cilia nor to analyze ciliary length. Better quality must be provided. How was the quantification done? And talking about ciliogenesis: to what extent does KO or OE of Intu and/or Stat1 affect the number of ciliated cells?”

Response: Original Fig. 7a (now Suppl Fig. 9) was taken at low magnification to show a whole picture about cilia and cell death. We have replaced Fig 7a with new images of higher magnification and better quality to show cilia. We have also provided more details about the quantification of cilium length in the Materials and Methods section (Page 17). We counted the number of ciliated cells in different experiment settings to show cilia frequency as an indication of ciliogenesis. Intu KO seemed to reduce cilia frequency (Fig 7b, 7c), whereas Intu OE increased it (Fig 7d). STAT1 KO slightly increased cilia frequency (Fig 7e).

General comments:

- In many western blots molecular weight standards are missing making it difficult to draw any conclusion out of the data (e.g. Fig 4b, Fig 3, ...).

We have now marked the positions of MW markers in all blots.

- Several western blots lack controls and the quantification does not always fit to the presented blot. E.g. the quantification of Parp/actin in Fig. 5e. This is not convincing. In any case, solid densitometric analyses might be nearly impossible in case bands are not clearly separated.

We have replaced the blots. Yes, we understand the difficulty of having reliable densitometric analysis.

- The manuscript requires some language editing (e.g. “Kidney tissues are characterized by primary cilia and PCP...” (pg 3) etc.

We have gone through the whole manuscript for language editing.

- “the site of ciliogenesis” is not the very best description for the basal body.

Revised.

- The authors should more clearly distinguish between “ciliogenesis” and “ciliary elongation”. If Intu is involved in cilia elongation ( longer cilia) , that would not necessarily mean it has a function in ciliogenesis, which typically refers to the very early step, the “birth” of this lovely organelle.

Yes, we fully agree. We have revised the description to distinguish between “ciliogenesis” and “ciliary elongation”. Cilia frequency was also measured to indicate ciliogenesis.

Reviewer 2

“This interesting manuscript proposes a new twist to a pathway that protects renal tubule cells from apoptosis during recovery from ischemia/reperfusion injury. The authors find that the CPLANE protein Intu, a component of the basal body previously proposed to load Ift proteins into cilia, regulates STAT1. After injury, STAT1 is activated to promote the apoptotic injury response. The authors suggest that Intu protects from this response by binding and promoting the proteasome dependent degradation of STAT1. In addition, tubule cells that recover from injury elongate their cilia, while those that do not display shorter cilia. While Intu is associated with ciliogenesis in other contexts, its absence in tubules has no effect on wildtype cilia length. However, Intu protects cilium length after injury, and here again this function is antagonistic with STAT1.

The story is important, and in principle I’m in favor of publication. However, There are several critical documentation issues that must first be addressed. In addition, several questions should be answered, and some issues with the introduction should be addressed.”

Response: We thank the reviewer for highly appreciating the significance and overall quality of the work. We have addressed the specific issues as summarized below.

“Definitions of precisely what is being quantified is missing for most graphs. For example, in Fig 2a, the number of apoptotic cells – what is the denominator? Similar definitions are needed for most of the subsequent graphs as well.”

Response: We have gone through all figure panels to add the denominators or relevant information to clarify the quantification data.

“The confocal images are of marginal quality, and I’m struggling to see how the authors can reliably quantify cilium length, at least from images like the ones provided. Cilia tend to bend and point down the long axis of the tubule, and thus their length is very difficult to assess except in longitudinal slices. Some description of how the measurements were made, along with high quality images, is needed.”

Response: We agree with the reviewer’s expert comments. Indeed, it is difficult to accurately measure cilia length in renal tubules in kidney tissues. Fortunately,

proximal tubules have much shorter cilia (and less bending) than other renal tubule segments, making it easier to measure in our study. As suggested, We have provided a more detailed description about the measurement in the Materials and Methods section (Page 17). We have also included better confocal images of kidney sections (Fig. 7).

“Along those lines, the authors show that cilium response is opposite in apoptotic and non-apoptotic cells. However, in subsequent experiments, the data are not divided between apoptotic and non-apoptotic cells. This seems to be an important distinction that may segregate the data in a meaningful way, and should be addressed.”

Response: As suggested, we have analyzed the cilium response separately in TUNEL-negative vs. TUNEL-positive cells in PT-Intu-WT and -KO mice, respectively (Suppl Fig. 10). The results are consistent with our conclusions: 1) apoptotic cells have shorter cilia, and 2) PT-Intu-KO kidney tubules have shorter cilia than WT in both apoptotic and non-apoptotic cells.

“Line 77: Intu was induced in WT kidneys but not in KO kidneys. Is there no expression in more distal nephron segments? Since the KO is specific to proximal tubules, one might expect to see some induction from the unaffected segments.”

Response: The induction of Intu in WT kidneys is a response to stress and injury. In renal ischemia-reperfusion, proximal tubules are the main injury site. Therefore, Intu induction occurs in mainly proximal tubules and not much in distal tubules.

“How do the authors explain that STAT1 is the major band on the Fig 4a TAP pulldown gel? What about the rest of the CPLANE complex? This is quite unexpected.”

Response: Excellent point. In addition to STAT1, our TAP assay pulled down many other proteins. We have now provided the full-lane silver staining gel image and also a table listing all the proteins detected by mass spectrometry in the TAP assay (Suppl Fig. 4 and Suppl Table 1).

“Do the authors agree that the subsequent pull down assays suggest a weak interaction with only a small fraction bound?”

This is possible. Technically, the pulldown assay is affected by many factors such as antibody affinity, antibody dilution and incubation time, and film exposure time etc.

“How do the authors explain the apparent cloud of STAT1 immunofluorescence near but not bound to the basal bodies?”

Response: We currently do not have a good explanation to the subcellular localization of total STAT1. These cytoplasmic STAT1 molecules may have a dynamic subcellular location associated with centrosome matrix and cell cycle

during cell proliferation. In differentiated epithelial cells, STAT1 may vertically distribute at the apical side around centrosome. We suspect that STAT1 may play a role in centrosome movement and replication. We also wonder whether STAT1, as an important protein in apoptosis regulation, is involved in cell polarity and planar polarization. Apparently, further studies need to address these possibilities.

“Line 46: “The PCP core proteins include those of the Frizzled and the Fat systems 1, 7.” In the PCP literature, the “core system” refers to proteins of the Fz, Vangl, Celsr, etc group, while the Fat system is not considered part of the core but a separate mechanism.”

We have revised the description, thanks.

Line 48 “Recently, PCP has been linked to primary cilia.” This is a complex and controversial area, and in general the idea that there is an intimate association is suspect. The authors should look at reviews by John Wallingford, and provide a more nuanced discussion.

We agree, the description has been revised.

Line 59 “Kidney tissues are characterized by primary cilia and PCP, and dysregulation of cilia and PCP contributes to the pathogenesis of renal diseases, such as polycystic kidney disease 19.” Similarly, this idea is complex, and while embedded in the literature, there is little direct evidence. Again, a more circumspect telling is warranted.

As suggested, we have revised the description.

“Finally, the authors fail to acknowledge prior work indicating that STAT1 is involved in the injury response. For example, Yang et al 2008 Jan;23(1):91-100, Si et al J Transl Med. 2013 Jun 9;11:141, and perhaps Tang et al AJN 2010;32(2):145-55 should be cited.”

Thanks for pointing out these articles, which have now been cited in our manuscript (new refs. 35-37).

Reviewer 3

“This study analyzes the role of the CPLANE protein, Intu, in recovery from renal ischemia-reperfusion injury. Intu is thought to have a role in ciliogenesis and planar cell polarity (PCP) and has been implicated in ciliopathy disorders in frog and humans. Here, Intu is ablated in the proximal tubule (PT), aggravating recovery from renal ischemia-reperfusion injury due to defective ciliogenesis. Interaction of Intu with STAT1 at the base of the cilium is proposed to regulate STAT1 degradation. Hence, a process of recovery via targeting STAT1 for degradation is proposed.”

We thank the reviewer for thoroughly analyzing the main findings in this study.

“1. At what age does the PEPCK-Cre inactivate Intu in the PT? Is it after completion of kidney development? Are any developmental defects found in these kidneys? Do these animals have PCP defects?”

Response: In mice, PEPCK-Cre starts to express for floxed gene ablation at ~3 weeks after birth when kidney development has mostly completed. Consistently, PT-Intu-KO mice in our study showed normal renal histology and function, without noticeable signs of kidney development defects or PCP defects.

“2. Since knockout mice are available, it seem strange that shRNA, with incomplete silencing, was used for the cellular analysis in Figure 2.”

Response: In addition to Figure 2, the cell culture model was used in subsequent experiments involving various manipulations such as single and double transfections of shRNAs as well as overexpression plasmids. We chose to use the cell culture model, because primary cells isolated from mice are technically much more difficult for these manipulations.

“3. Further data about the TAP study should be shown. Were other interacting proteins identified? How do they compare to the proteins identified by Toriyama et al?”

Response: Our TAP assay also pulled down other proteins. We have now provided the image of the full-lane silver staining gel to show the major bands and the list of pulled down protein identified by mass spectrometry (Suppl Fig. 4, Suppl Table 1). We revealed many identical CPLANE interactome proteins that were originally reported by Toriyama et al.

4. Interaction by co-IP between endogenous Intu and STAT1 should be shown. In addition, the role of Intu in targeting STAT1 for degradation should be analyzing by loss of the endogenous protein.

Response: We have performed co-IP between endogenous Intu and STAT1 in WT and KO mice kidneys (Fig. 4d). We have also shown STAT1 increase in Intu-KO mice kidneys (Fig. 5e).

“5. The cilium length differences in Figure 7 seem marginal, and measuring cilia length differences in cultured cells is fairly unreliable. Data from STAT1 knockout mice would strengthen this analysis.”

Response: We have repeated some of the cilia measurements. We did realize that the orientation of cilia affects the measurement accuracy. Thus, we did our best to measure the length accurately and counted more cilia in each experiment. We understand that data from STAT1-KO mice would strengthen the results. Unfortunately, we currently do not have STAT1-KO mice. Moreover, analysis of STAT1-KO mice seems beyond the scope of the current study that is focused on the role of Intu in ischemia-reperfusion kidney injury and the underlying

mechanisms. In addition, several studies have reported the role of STAT1 in renal ischemia-reperfusion injury (Yang N. Nephrol Dial Transplant 2008; Tang S. Am J Nephrol 2010; Si Y. J Transl Med 2013). We have indicated this in page 8.

“6. The study would be more complete if the information found from the Intu/STAT1 interaction could be translated back into a treatment option that is shown to be of value for treatment of ischemia-reperfusion injury in vivo.”

Yes, it would be great if we can target it for therapy. This apparently depends on the development or discovery of specific pharmacological agents to modulate Intu-STAT1 interaction. We have added a brief discussion of this possibility (page 9).

Finally, we would once again thank the reviewers for all the thoughtful comments and suggestions, which have guided the revision and helped improve our study.

REVIEWERS' COMMENTS:

Reviewer #1 (Remarks to the Author):

The authors addressed all my concerns in their revised manuscript.

Only one minor point: The Tunel staining should be presented within Fig 7a to demonstrate negative or positive TUNEL.

Reviewer #2 (Remarks to the Author):

The authors have responded well to most of the prior criticisms, and the manuscript is acceptable for publication with several further modest revisions.

"Line 53 Almost every kidney tubular cell has a primary cilium protruding towards the lumen and these cells are arranged in a planar polarization pattern."

No planar polarized pattern of cell arrangement has been reported. Convergent extension during development has been inferred, but is not visible in fixed sections nor would it be evident in mature tubules.

"Line 80 To further functionally characterize Intu, we examined the effect of stable Intu knockdown with specific shRNA (shIntu; Fig. 2b, c)."

Indicate that this is in BUMPT cells.

"Line 141 known to contribute to ciliogenesis 27. After demonstrating the physical and functional interactions between Intu and STAT1 during cell stress, we wondered if STAT1 participates in Intu-mediated ciliogenesis. We initially analyzed the association of apoptosis and cilium length in kidney tubules after renal I/R injury in WT mice, showing that cilia grew longer in TUNEL negative/non-apoptotic tubules, and diminished in TUNEL-positive/apoptotic tubules (Fig. 7a, Supplementary Fig. 9)."

As noted by another reviewer, ciliogenesis is not equivalent to cilium length. If one believes the measurements, there is a difference in length. The frequency of cilia may represent an effect on ciliogenesis, cilium loss, or both, but this cannot be determined. Therefore, please change the phrase "we wondered if STAT1 participates in Intu-mediated ciliogenesis" to indicate examination of length.

"Line 345 cilium length and quantification of cilium frequency. Briefly, cilium length was shown in collected images by drawing a line along the primary cilia. If cilium length was too short, images would be expanded so that the length was measurable. To maximize the accuracy, more than 50 cilia were measured in each condition."

This description is still inadequate – how do the authors know the whole cilium is imaged? How do they account for curvature out of the image plane? What is the specimen thickness? Measuring a projection of a 3D curved structure in a 2D image will introduce error. This error may be similar among different measurements to still show meaningful differences.

Reviewer #3 (Remarks to the Author):

The authors have been responsive to my earlier comments and have either added additional data to address the issue or discussed the issue in the paper.

I have no further comments.

Point-by-point Response to Reviewers

Reviewer #1

The authors addressed all my concerns in their revised manuscript.

Only one minor point: The TUNEL staining should be presented within Fig 7a to demonstrate negative or positive TUNEL.

As suggested, we have added the TUNEL-negative and –positive images in Fig 7a.

Reviewer #2

The authors have responded well to most of the prior criticisms, and the manuscript is acceptable for publication with several further modest revisions.

“Line 53 Almost every kidney tubular cell has a primary cilium protruding towards the lumen and these cells are arranged in a planar polarization pattern.” No planar polarized pattern of cell arrangement has been reported. Convergent extension during development has been inferred, but is not visible in fixed sections nor would it be evident in mature tubules.

“and these cells are arranged in a planar polarization pattern” has been deleted.

“Line 80 To further functionally characterize Intu, we examined the effect of stable Intu knockdown with specific shRNA (shIntu; Fig. 2b, c).” Indicate that this is in BUMPT cells.

“in BUMPT cells” has been added to specify.

“Line 141 known to contribute to ciliogenesis 27. After demonstrating the physical and functional interactions between Intu and STAT1 during cell stress, we wondered if STAT1 participates in Intu-mediated ciliogenesis. We initially analyzed the association of apoptosis and cilium length in kidney tubules after renal I/R injury in WT mice, showing that cilia grew longer in TUNEL negative/non-apoptotic tubules, and diminished in TUNEL-positive/apoptotic tubules (Fig. 7a, Supplementary Fig. 9).” As noted by another reviewer, ciliogenesis is not equivalent to cilium length. If one believes the measurements, there is a difference in length. The frequency of cilia may represent an effect on ciliogenesis, cilium loss, or both, but this cannot be determined. Therefore, please change the phrase “we wondered if STAT1 participates in Intu-mediated ciliogenesis” to indicate examination of length.

Excellent point. It has been changes to “we wondered if STAT1 participates in Intu-mediated cilium length regulation”

“Line 345 cilium length and quantification of cilium frequency. Briefly, cilium length was shown in collected images by drawing a line along the primary cilia. If cilium length was too short, images would be expanded so that the length was measurable. To maximize the accuracy, more than 50 cilia were measured in each condition.” This description is still inadequate – how do the authors know the whole cilium is imaged? How do they account for curvature out of the image plane? What is the specimen thickness? Measuring a projection of a 3D curved structure in a 2D image will introduce error. This error may be similar among different measurements to still show meaningful differences.

We understand. We have added a clarification, “Due to the curvature of primary cilia, we only measured the visible cilium length on 4 μm kidney sections.”

Reviewer #3

The authors have been responsive to my earlier comments and have either added additional data to address the issue or discussed the issue in the paper.

I have no further comments.